# TodoEvolve: Learning to Architect Agent Planning Systems

Jiaxi Liu [*1]   Guibin Zhang [*†2]   Yanzuo Jiang [1]   Zihan Zhang [34]   Heng Chang [5]
Zhenfei Yin [‡6]   Qibing Ren [‡1]   Junchi Yan [‡14]

## Abstract

Planning has become a central capability for contemporary agent systems in navigating complex, long-horizon tasks, yet existing approaches predominantly rely on fixed, hand-crafted planning structures that lack the flexibility to adapt to the structural diversity of open-ended problems. To address this limitation, we introduce `TodoEvolve`, a meta-planning paradigm that autonomously synthesizes and dynamically revises task-specific planning architectures. Specifically, we first construct `PlanFactory`, a modular design space that standardizes diverse planning paradigms within a unified codebase encompassing topology, initialization, adaptation, and navigation, thereby providing a common interface for heterogeneous planning patterns. Leveraging `PlanFactory`, we collect high-quality planning trajectories and train `Todo-14B` via *Impedance-Guided Preference Optimization* (IGPO), a multi-objective reinforcement learning objective that encourages the generation of planning systems that are performant, stable, and token-efficient across arbitrary tasks and agent backbones. Empirical evaluations on five agentic benchmarks demonstrate that `TodoEvolve` consistently surpasses carefully engineered planning modules while maintaining economical API costs and runtime overhead. Our codes are available at https://github.com/EcthelionLiu/TodoEvolve.

*Equal contribution   † Project leader.   [1]Shanghai Jiao Tong University [2]National University of Singapore [3]NanKai University [4]Shanghai Innovation Institute [5]Tsinghua University [6]Oxford University. Correspondence to: Zhenfei Yin <jeremyyin@robots.ox.ac.uk>, Qibing Ren <renqibing@sjtu.edu.cn>, Junchi Yan <yanjunchi@sjtu.edu.cn>.

*Proceedings of the $43^{rd}$ International Conference on Machine Learning*, Seoul, South Korea. PMLR 306, 2026. Copyright 2026 by the author(s).

## 1. Introduction

With the rapid advancement of foundation models (Team et al., 2025a; GLM Team, 2025; Team et al., 2025b), large language model (LLM)-powered agents have begun to demonstrate strong capabilities across domains such as deep research (Hu et al., 2025a; Shi et al., 2025b), complex software engineering (iQuest, 2025; Yang et al., 2024), and real-world transactions (Andon, 2025; Backlund & Petersson, 2025). Beyond improvements in base model capacity, increasingly sophisticated agent scaffolds are equally critical (Wang et al., 2024b), equipping LLMs with essential agentic support including planning (Wu et al., 2025a), memory (Hu et al., 2026a), reflection, *etc*. Among these, planning stands out as a central capability, enabling agents to navigate complex environments by maintaining a coherent global state, preserving behavioral consistency, and coordinating actions across tasks (Cao et al., 2025).

Existing planning systems developed for LLM-based agents exhibit substantial diversity. From the perspective of **planning target**, some are designed to support *single agent*, primarily addressing long-horizon execution and mitigating the risk of "*lost in the middle*" (Erdogan et al., 2025), while others are tailored for *multi-agent systems*, focusing on subtask allocation and contextual coordination across agents with distinct roles (Parmar et al., 2025; Hu et al., 2025b). In terms of **representational form**, plans have been instantiated using a wide range of structures, including linear to-do lists (LangChain, 2025), directed acyclic graphs (DAG) (Qin et al., 2025), tree-structured plans (Hu et al., 2026b), and hierarchical notes. Moreover, planning systems differ markedly across **task domains**, with domain-specific designs emerging for embodied action (Wang et al., 2023), web search (Kim et al., 2024), and programming. Faced with this diversity, practitioners may naturally ask: *is there a single planning structure that can serve as a one-size-fits-all solution that generalizes well across settings?*

We posit that such an oracle planning system does not exist. Beyond distinct task domains require different planning priors (for instance, MCTS-based planning may be effective for mathematical reasoning yet is rarely adopted for autonomous driving agents due to the vastness of its action space (Wang et al., 2024a)), even within a single task

class, alternative planning priors exhibit performance disparities. For example, in web search, AOP (Li et al., 2025a) employs a simple linear to-do list coupled with a reward model to solve document QA in a token-efficient manner, but it is substantially outperformed in more complex multimodal settings by DAG-based planning structures (Qin et al., 2025). Similarly, while linear tasks require minimal revision (Hu et al., 2025b), high-conflict environments demand continuous topological restructuring (Zhang et al., 2025a), rendering a single, universal planning system unrealistic.

Accordingly, we contend that the central challenge is not to design a *one-size-fits-all* planner, but to *customize* planning systems to the structural characteristics of each task. To this end, we propose `TodoEvolve`, a meta-planning paradigm that synthesizes task-adaptive agentic planners and dynamically updates their planning states as execution unfolds. Concretely, we train `Todo-14B` using *Impedance-Guided Preference Optimization* (IGPO), a multi-objective preference learning objective that jointly promotes high performance, stability, and token efficiency in the generated planning systems. The resulting meta-planner `Todo-14B` takes a task instance as input and instantiates a tailored planning topology, revision cadence, and navigation strategy, operationalized as a task-specific *to-do* structure. `Todo-14B` integrates seamlessly with single/multi-agent execution frameworks, remains compatible with diverse LLM backbones, and generalizes across heterogeneous task domains.

To ground `TodoEvolve` within the diverse landscape of existing planning systems, we introduce a modular planning design space comprising four dimensions: ♣ **Topology** (the structural organization of task decomposition), ♦ **Initialization** (how the task topology is instantiated), ♥ **Adaptation** (when and how the topology is revised), and ♠ **Navigation** (the mechanism that issues executable directives to the acting agent). This design space provides a unified abstraction capable of accommodating and localizing a wide spectrum of existing planning paradigms. Building on this formulation, we decompose and re-implement seven representative planning architectures, including Plan-and-Act (Erdogan et al., 2025), linear planning (Hu et al., 2025b), DAG-based planning (Qin et al., 2025), and parallel and dynamic planning (Zhu et al., 2025). The resulting framework, denoted as `PlanFactory`, serves both as **(i) a data synthesis engine** for generating high-quality planning trajectories to train `TodoEvolve` and **(ii) a standardized codebase** to facilitate future research on agentic planning capabilities. Our contributions are as follows:

❶ **Unified Codebase:** We introduce `PlanFactory`, a modular design space for agentic planning systems encompassing four key components (*topology*, *initialization*, *adaptation*, and *navigation*), providing unified implementations and benchmark support for a wide range of prevailing planning structures.

❷ **Meta Planners:** We introduce `TodoEvolve`, a meta-planning paradigm that synthesizes task-adaptive planning systems and dynamically revises planning states. Through impedance-guided preference optimization (IGPO), we train `Todo-14B`, a meta-planner capable of instantiating and controlling planning structures across diverse scenarios and agent backbones.

❸ **Experimental Evaluation:** Extensive experiments on five challenging agentic benchmarks demonstrate that `TodoEvolve` delivers (I) **substantial performance gains**, improving frameworks such as Smolagents by up to 16.37% on GAIA; and (II) **robust generalization**, generalizing across diverse LLM backbones, for example boosting GPT-5-Mini to 75% on xBench-DS.

## 2. Related Works

**Agent Planning Systems.** Agentic planning has evolved from static prompting to structured reasoning. Foundational works like CoT (Wei et al., 2022), ToT (Yao et al., 2023a), and GoT (Besta et al., 2024) enabled cognitive decomposition, while ReAct (Yao et al., 2023b) and Reflexion (Shinn et al., 2023) introduced execution loops with self-correction. However, these approaches typically rely on rigid, predetermined topologies, limiting adaptability in open-ended environments where optimal structures vary dynamically. Recent frameworks embed domain priors: FlashSearcher (Qin et al., 2025) and OAgents (Zhu et al., 2025) leverage DAG-based parallelism; OWL (Hu et al., 2025b) and AgentOrchestra (Zhang et al., 2025b) utilize hierarchical coordination; and systems like FlowSearch (Hu et al., 2026b), JoyAgent (Liu et al., 2025), and Co-Sight (Zhang et al., 2025a) optimize workflows via structured verification. Crucially, these systems remain bound by *pre-designed* architectures. This necessitates a meta-planning approach capable of autonomously synthesizing and customizing planning structures tailored to each task's unique complexity.

**RL for Agent Planning.** Training paradigms have shifted from preference alignment (e.g., DPO (Rafailov et al., 2023), PPO (Schulman et al., 2017)) toward reinforcement learning with verifiable rewards (RLVR). As exemplified by DeepSeek-R1 (Guo et al., 2025), optimizing against objective ground truths fosters emergent self-verification, shifting focus from human style to logical correctness. Recent works apply this to diverse dimensions: Search-R1 (Jin et al., 2025) and LATS (Zhou et al., 2023) optimize search trajectories; RAGEN (Wang et al., 2025) targets multi-turn interactions; and ToRL (Li et al., 2025b) refines tool-use strategies. However, a critical limitation persists: these approaches primarily optimize the agent's action policy or tool selection within fixed topological loops. In contrast, our work leverages verifiable trajectories to train a meta-planner, moving beyond policy optimization to autonomously synthesize the underlying planning structure itself.

*Table 1.* An overview of agentic planning paradigms decomposed in `PlanFactory`. The "Mul" column distinguishes between single-agent ($\mathcal{S}$) and multi-agent ($\mathcal{M}$) compatibility. "Scope" specifies the granularity at which planning is performed ($\alpha$ for step-wise vs. $\Omega$ for task-wise), and "Struct" indicates whether the execution flow is linear ($\ell$) or organized as a complex graph structure ($\mathcal{G}$).

| Method | Date | Mul. $(\mathcal{M}/\mathcal{S})$ | Scope $(\Omega/\alpha)$ | Struct. $(\mathcal{G}/\ell)$ | ♣ Topology *Structural Organization* | ♦ Initialization *Instantiation Mechanism* | ♥ Adaptation *Revision Logic* | ♠ Navigation *Execution Directives* |
|---|---|---|---|---|---|---|---|---|
| **OWL** | 2025.6 | $\mathcal{M}$ | $\Omega$ | $\mathcal{G}$ | **Dual Hierarchy** | **Planner Decompose** | **Manager Intervention** | **Dynamic Dispatch** |
| **OAgents** | 2025.6 | $\mathcal{M}$ | $\alpha$ | $\ell$ | **Modular Graph** | **SOP Configuration** | **Critic-Loop Feedback** | **Loop Execution** |
| **AgentOrchestra** | 2025.9 | $\mathcal{M}$ | $\Omega$ | $\mathcal{G}$ | **Orch. Hierarchy** | **Role Definition** | **Env Feedback** | **Centralized Routing** |
| **Flash-Searcher** | 2025.9 | $\mathcal{S}$ | $\Omega$ | $\mathcal{G}$ | **Parallel DAG** | **Dependency Parsing** | **Workflow Pruning** | **Concurrent Paths** |
| **JoyAgent** | 2025.10 | $\mathcal{M}$ | $\Omega$ | $\mathcal{G}$ | **Collective Hierarchy** | **Hybrid Planning** | **Consensus Voting** | **Joint Deliberation** |
| **FlowSearch** | 2025.10 | $\mathcal{M}$ | $\Omega$ | $\mathcal{G}$ | **Thought Graph** | **Flow Construction** | **Dynamic Expansion** | **Graph Traversal** |
| **Co-Sight** | 2025.10 | $\mathcal{M}$ | $\alpha$ | $\ell$ | **Cross-Check Net** | **Inconsistency Trigger** | **Meta-Verification** | **Conflict Resolution** |

## 3. `PlanFactory`: Unified Planning Codebase

### 3.1. Preliminary

We adopt a bi-level agentic inference abstraction where the Agent System executes environment interactions, while the Planning System governs high-level control logic.

**Agent Systems.** We formalize the execution substrate as a tuple $\mathcal{M} = \langle I, \mathcal{S}, \mathcal{A}, \Psi, \Omega \rangle$, comprising an agent roster $I$, a global state space $\mathcal{S}$, and a joint action space $\mathcal{A} = \bigcup_{i \in I} \mathcal{A}_i$. The state dynamics follow $\Psi(s_{t+1} \mid s_t, a_t, \mu(t))$, where $\mu(t) \in I$ identifies the active agent at time $t$. To support action generation, a context mechanism $\Omega$ aggregates the execution history $H_t$, such that $a_t = \pi_{\mu(t)}(s_t, H_t, Q \mid \Omega)$. Finally, the resulting trajectory $\tau$ is evaluated by a reward $R(\tau)$, positioning $\mathcal{M}$ as a flexible execution engine orchestrated by higher-level logic.

**Planning Systems.** The Planning System imposes structural logic on execution. We formalize it as a configuration $\mathcal{P}$ comprising four key functional modules:

$$\mathcal{P} = \langle \mathcal{G}, \mathcal{I}_{init}, \mathcal{F}_{adapt}, \mathcal{N}_{nav} \rangle \tag{1}$$

defining the mechanisms respectively. As shown in Table 1, existing paradigms represent static instances of $\mathcal{P}$, augmenting the policy as $a_t = \pi(\cdot \mid \mathcal{P})$. Crucially, current systems rely on manual engineering to fix $\mathcal{P}$, limiting adaptability. This motivates our meta-level framework, which automatically synthesizes an optimal $\mathcal{P}^*$ tailored to each task.

### 3.2. PlanFactory Codebase

We present `PlanFactory`, a modular toolkit designed to decouple high-level planning logic from low-level execution, facilitating the systematic study of agentic architectures.

**Implementation.** The core of PlanFactory is a standardized lifecycle interface. All planning paradigms (Table 1) inherit from the `BasePlanning` abstract class, which encapsulates the four essential components: ♣ **Topology**, ♦ **Initialization**, ♥ **Adaptation**, and ♠ **Navigation**. For more details, please refer to Appendix A. This polymorphism allows heterogeneous strategies to be swapped seamlessly within a shared runtime. Crucially, this design supports highly parallelized inference, enabling users to benchmark disparate configurations concurrently on a unified backend without refactoring the agent loop.

**Evaluation.** PlanFactory provides a comprehensive evaluation suite tailored for dynamic information-seeking tasks. To ensure reliable assessment in open domains, we employ an LLM-as-a-Judge mechanism. This automates trajectory analysis, rigorously quantifying both task success rates and the logical coherence of the generated plans.

## 4. `TodoEvolve`: Training Meta-Planners

Current agentic systems predominantly rely on static protocols, which inherently lack the flexibility to address the diverse distribution of real-world queries. To break the shackles of manual engineering, we propose a Generative Planning Paradigm. The core of this paradigm is **Impedance-Guided Preference Optimization (IGPO)**, a novel training strategy designed to endow `Todo-14B` with the ability to dynamically synthesize bespoke planning systems $\mathcal{P}_{custom}$ tailored to unique structural requirements. Unlike standard alignment which focuses on stylistic imitation, IGPO explicitly optimizes the meta-planner to maximize execution stability while minimizing computational overhead. This section elaborates on our dual-track methodology: (I) constructing a high-quality verifiable planning dataset, and (II) employing IGPO to establish robust architectural reasoning.

### 4.1. Data Construction

To enable generative planning, we formulate the system design as a conditional code generation task. To bridge the lack of architectural priors in standard LLMs, we propose a *Bootstrap-and-Filter* pipeline within `PlanFactory` that transforms the search for optimal plans into a high-quality supervised dataset. This process involves four stages:

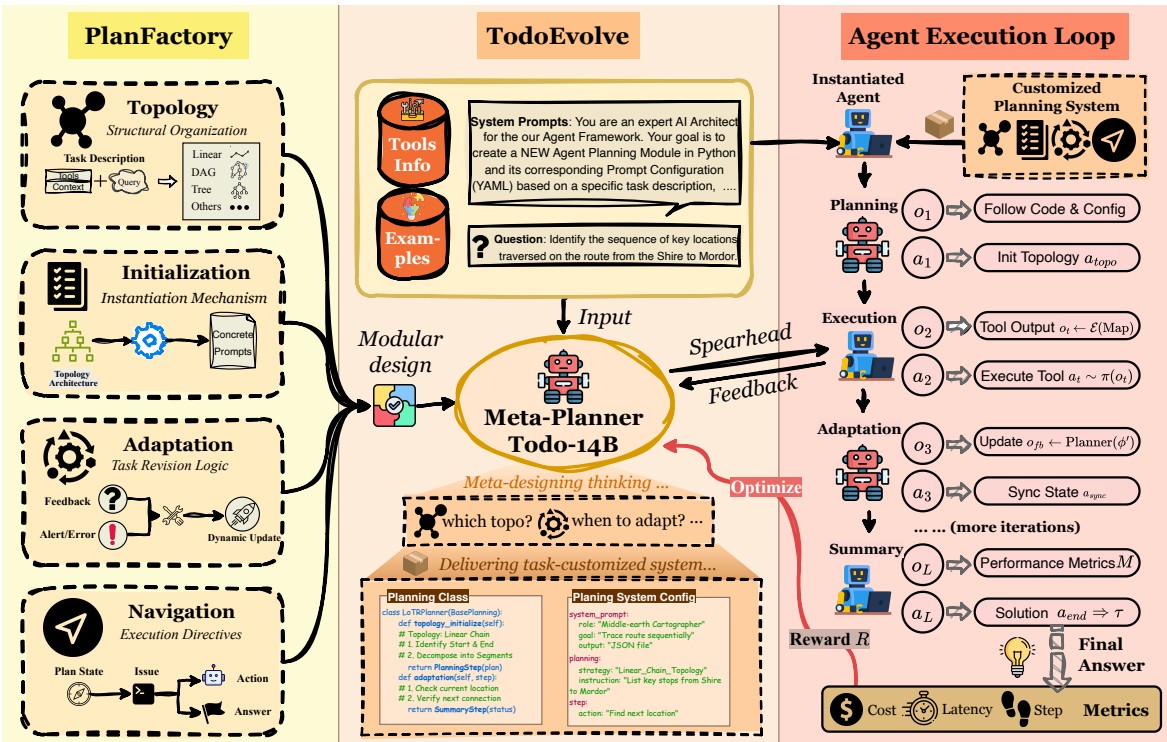

**Figure 1.** The overall inference workflow of `TodoEvolve` first constructs a customized planning system along four dimensions—topology, initialization, adaptation, and navigation, and then deploys it in real time to orchestrate agent execution.

**Phase 1: Standardization via Unified Tool Interface.** First, we utilize the modular nature of `PlanFactory` to deconstruct the functional primitives of existing representative planning systems, specifically the 7 paradigms listed in Table 1. We decompose their discrete mechanisms into standardized tools. These tools are encapsulated within our unified framework, creating a shared *Plan Space* where different topological structures can be expressed using a consistent code interface.

**Phase 2: Evolutionary Sampling.** With the standardized tools ready, we employ an evolutionary strategy to generate diverse planning candidates. For each query $Q_i$, we construct a specialized input context $\mathcal{C}_i$ consisting of:

- The specific user query $Q_i$.
- The system prompt defining the Meta-Planner's role.
- Detailed documentation of the available Meta-Tools.
- A randomly sampled subset of 3 static planning samples $\{P_{ref}^1, P_{ref}^2, P_{ref}^3\}$ from our standardized pool, serving as structural references to guide the architectural design.

The model is tasked with synthesizing an intermediate reasoning trace enclosed , followed by a unique, query-specific plan $P_{gen}$ that integrates or modifies the sampled patterns to best suit $Q_i$. This process encourages the model to reason about architectural choices before committing to a final planning structure, thereby promoting task-specific adaptation rather than simple template replication.

**Phase 3: Execution-Based Verification.** We validate each synthesized plan $P_{gen}$ by executing it within the Plan-Factory runtime to generate a trajectory $\tau$ and final answer $A_{final}$. We apply a strict Execution-as-Judge filter: $P_{gen}$ is retained into the dataset if and only if $A_{final}$ matches the ground truth. This mechanism effectively purges hallucinated or unsound architectures, ensuring the Meta-Planner learns exclusively from successful design patterns.

**Phase 4: Preference Construction for SFT and IGPO.** Finally, we format the validated execution trajectories into training supervision. To instill both correctness and efficiency into the Meta-Planner, we employ a dual-track alignment strategy, that separates fundamental capability learning from preference-based refinement:

**SFT Data Construction:** During SFT, we adopt a strict outcome-supervised filtering protocol. We iterate through the generated plan candidates and retain only those pairs $(\mathcal{C}_i, P_{gen})$ that successfully execute. By grounding the target plan $P_{gen}$ on the reference-augmented context $\mathcal{C}_i$, we ensure that the base model learns to synthesize valid, executable architectures from the provided structural inspirations.

**IGPO Data Construction:** To further align the model with high-quality planning logic via process supervision, we construct preference pairs $(P_{win}, P_{lose})$ for IGPO. We process the sampling results in pairs and determine the winner using a hierarchical criterion:

- **Correctness First:** Correctness is the prerequisite. If one plan succeeds and the other fails, the successful plan is strictly preferred ($P_{win} \succ P_{lose}$).
- **Noise Filtering:** Pairs where both failed are discarded.
- **Efficiency as Tie-Breaker:** In "expert scenarios" where both candidates yield correct answers, we introduce a novel metric, **Cognitive Impedance** ($\mathcal{I}$), to resolve the tie. We define $\mathcal{I}$ as a compound cost function:

$$\mathcal{I}(\tau) = C_{tot} \cdot \exp\left(\lambda_1 N_{fail} + \lambda_2(1 - S_{stab}) + \lambda_3 \frac{C_{plan}}{C_{exec}}\right) \quad (2)$$

where $C_{tot}$ is the total cost, $N_{fail}$ counts errors, and $S_{stab}$ quantifies execution smoothness. Crucially, the ratio of planning cost ($C_{plan}$) to execution cost ($C_{exec}$) acts as a bureaucracy penalty, ensuring planning effort does not outweigh execution.

Formally, this pipeline yields two corpora: $\mathcal{D}_{SFT} = \{(\mathcal{C}_i, P_{gen}) \mid \text{Correct}(P_{gen})\}$ for structural competence, and $\mathcal{D}_{IGPO} = \{(\mathcal{C}_i, P_{win}, P_{lose}) \mid P_{win} \succ P_{lose}\}$ for efficiency alignment.

### 4.2. `Todo-14B`: Training Meta-Planner

This section details the training methodology for `Todo-14B`. We optimize the Meta-Planner $\pi_\theta$ to synthesize planning configurations that maximize downstream agent performance. We adopt a two-stage curriculum: SFT establishes structural competence, followed by IGPO to align the planner with execution efficiency.

#### 4.2.1. STAGE 1: STRUCTURAL COMPETENCE VIA SFT

We first instill the fundamental capabilities of code generation and architectural reasoning into the Meta-Planner. Leveraging $\mathcal{D}_{SFT}$, we treat the verified pairs $(\mathcal{C}, Pgen)$ as expert demonstrations. We optimize $\pi_\theta$ using the standard next-token prediction objective by minimizing the negative log-likelihood of the target sequence. This supervised training serves as a crucial warm-start phase, ensuring that the model acquires the necessary syntactic rules and API constraints. Consequently, it learns to synthesize valid instances of $P$ that are structurally grounded in the context $\mathcal{C}$, providing a stable initialization for subsequent alignment.

#### 4.2.2. STAGE 2: IMPEDANCE-GUIDED PREFERENCE ALIGNMENT

While SFT ensures syntactic viability, it does not guarantee execution efficiency. The subspace of functionally correct plans is vast, yet the subset of optimal configurations—those that minimize resource consumption while maximizing success—is sparse. To transition from static correctness to dynamic optimality, we formulate planning generation as a meta-level optimization problem.

Let $P \in \mathcal{P}$ denote an executable plan configuration. The

*Table 2.* Detailed statistics of the constructed datasets. We operate in a long-context regime, where the input $L_{Context}$ ($\sim$13k tokens) is a composite sequence comprising the system prompt, tool definitions, retrieved structural examples, and the specific user query.

| Dataset Stage | Samples | Input ($L_{Context}$) | Reasoning ($L_{CoT}$) | Code ($L_{Code}$) |
|---|---|---|---|---|
| **Stage 1: SFT** | 3360 | $\sim$ 13,199 | $\sim$ 423 | $\sim$ 1,642 |
| **Stage 2: IGPO** | 2000 | $\sim$ 13,168 | $\sim$ 497 | $\sim$ 1,636 |

Meta-Planner searches the plan space for an optimal configuration $P^*$ that maximizes the expected return, balancing task success against operational costs:

$$P^* = \arg\max_{P \in \mathcal{P}} \mathbb{E}_{\tau \sim \mathcal{M}(P)}[R(\tau) - \lambda \mathcal{I}(\tau)] \quad (3)$$

where $R(\tau)$ is the binary success reward and $\mathcal{I}(\tau)$ represents the cognitive impedance. To solve this, we employ our IGPO method.

**Impedance-Contrastive Rejection Sampling.** Unlike standard preference collection which often relies on subjective ranking, our framework constructs preference pairs based on objective execution metrics. The data curation process functions as a rejection sampling mechanism designed to distill efficiency signals from stochastic exploration:

- **Exploratory Synthesis:** Given a context $\mathcal{C}$, the current policy $\pi_\theta$ samples $K$ candidate plans $\{\phi_1, \ldots, \phi_K\}$, inducing diverse transition dynamics for the agent system.
- **Execution & Evaluation:** The Agent System executes these plans to generate trajectories $\tau_i$. Each trajectory is evaluated using the composite impedance metric $\mathcal{I}(\tau_i)$, aggregating token consumption, temporal latency, and runtime errors.
- **Contrastive Pair Construction:** We construct the preference dataset $\mathcal{D}_{IGPO}$ by selecting pairs $(\phi_{win}, \phi_{lose})$. To ensure functional validity, we enforce $R(\tau_{win}) = 1$. A pair is selected only if there exists a significant impedance gap $\mathcal{I}(\tau_{lose}) - \mathcal{I}(\tau_{win}) > \delta$, ensuring the optimization is driven by high-confidence efficiency signals.

**Implicit Reward Alignment.** We posit that the optimal policy $\pi^*$ should assign probability mass to a configuration $\phi$ inversely proportional to its impedance, subject to a KL-divergence constraint that prevents deviation from the reference distribution. Defining the implicit reward as $r(\phi) = -\mathbb{E}[\mathcal{I}(\tau)]$ for successful trajectories, the optimal policy follows the Boltzmann distribution:

$$\pi^*(\phi \mid \mathcal{C}) \propto \pi_{ref}(\phi \mid \mathcal{C}) \cdot \exp\left(\frac{1}{\beta} r(\phi)\right) \quad (4)$$

This formulation allows us to bypass training an explicit reward model. Following the DPO derivation, the implicit reward $r_\theta(\phi)$ can be re-parameterized by the log-ratio of the policy likelihoods:

$$r_\theta(\phi) = \beta \log \frac{\pi_\theta(\phi \mid \mathcal{C})}{\pi_{ref}(\phi \mid \mathcal{C})} \quad (5)$$

*Table 3.* Performance of various agent frameworks on the WebWalkerQA, xBench-DS, TaskCraft, and GAIA benchmarks. For each column, the best and second-best pass@1 scores are highlighted in bold and underlined respectively.

| Framework | Model Family | WebWalker QA | xBench -DS | Task Craft | GAIA | | | |
|---|---|---|---|---|---|---|---|---|
| | | | | | Avg. | Level 1 | Level 2 | Level 3 |
| OWL Workforce pass@3 | GPT-4o+o3-mini | 57.64 | 55.0 | 58.33 | 60.61 | 81.14 | 58.14 | 26.92 |
| OWL RP pass@3 | GPT-4o+o3-mini | - | - | - | 58.18 | 81.14 | 54.65 | 23.08 |
| TapeAgents | Claude 3.7 etc. | - | - | - | 55.76 | 71.70 | 53.49 | 30.77 |
| AutoAgent | Claude 3.5 etc. | - | - | - | 55.15 | 71.70 | 53.40 | 26.92 |
| Smolagents | GPT-4.1 | - | - | - | 55.15 | 67.92 | 53.49 | 34.62 |
| Smolagents | GPT-5-mini | 58.82 | 51.0 | 64.00 | 55.75 | 69.81 | 54.65 | 30.77 |
| Magnetic-1 | OpenAI o1 etc. | - | - | - | 46.06 | 56.60 | 46.51 | 23.08 |
| Cognitive Kernel-Pro | Claude-3.7 etc. | 60.64 | 56.0 | 66.00 | 60.00 | 79.25 | 56.98 | 30.77 |
| Cognitive Kernel-Pro pass@3 | Claude-3.7 etc. | - | - | - | 75.15 | 84.91 | 73.26 | 61.54 |
| OAgents | Claude-3.7 etc. | 58.23 | 47.0 | - | 66.67 | 77.36 | 66.28 | 46.15 |
| Agent KB | GPT-4.1 | 60.59 | 48.0 | 61.67 | 61.21 | 79.25 | 58.14 | 34.62 |
| Agent KB pass@2 | GPT-4.1 | 68.82 | 58.0 | 72.67 | 67.27 | 83.02 | 67.44 | 34.62 |
| Agent KB pass@3 | GPT-4.1 | 73.53 | 68.0 | 75.33 | 73.94 | 84.91 | 73.26 | 53.85 |
| Flash-Searcher | GPT-5-mini | 71.18 | 69.0 | 69.67 | 69.09 | 79.25 | 69.77 | 46.15 |
| Flash-Searcher | Kimi K2 | 52.35 | 66.0 | 58.00 | 52.12 | 58.49 | 52.33 | 34.62 |
| Flash-Searcher | DeepSeek V3.2 | 69.41 | 68.0 | 69.33 | 60.61 | 79.25 | 53.49 | 46.15 |
| **TodoEvolve** + Smolagents | GPT-5-Mini | **73.53** | **75.0** | **72.67** | **72.12** | 81.14 | **72.09** | 46.15 |
| **TodoEvolve** + Smolagents | Kimi K2 | 64.71 | 71.0 | 69.33 | 60.00 | 73.58 | 55.81 | 46.15 |
| **TodoEvolve** + Smolagents | DeepSeek V3.2 | 70.59 | 74.0 | 71.33 | 70.91 | 84.91 | 67.44 | **53.85** |

The final IGPO loss function maximizes the margin between efficient and inefficient architectures by minimizing:

$$\mathcal{L}_{IGPO}(\theta) = -\mathbb{E}_{(\phi_w, \phi_l) \sim \mathcal{D}_{IGPO}} \left[ \log \sigma \left( r_\theta(\phi_w) - r_\theta(\phi_l) \right) \right] \quad (6)$$

This approach directly aligns the Meta-Planner with the execution environment, teaching it to architect systems that minimize cognitive impedance while maintaining functional correctness.

## 5. Experiments

### 5.1. Experiment Setup

**Training.** To equip our model with robust planning capabilities, we construct a high-quality composite dataset from diverse domains. Our training corpus aggregates samples from TaskCraft (Shi et al., 2025a), MoNaCo (Wolfson et al., 2026), WebWalkerQA (Wu et al., 2025b), and DeepSearchQA (Google DeepMind, 2025). The data construction pipeline uses a teacher-student paradigm, utilizing Gemini-3-Flash as the expert planner to generate high-level reasoning traces, and DeepSeek V3.2 as the executor to verify actionable outcomes. The curated dataset is detailed in Table 2. We employ Qwen3-14B (Yang et al., 2025) as our backbone model. This choice is motivated by its superior balance of reasoning density and instruction-following capabilities, making it an efficient yet powerful foundation for agentic tasks. The LLMs listed in 3, including GPT-5-mini, Kimi-K2, and DeepSeek V3.2, serve as downstream executor backbones. Training uses 8× NVIDIA H200 GPUs.

**Testing & Baselines.** To rigorously evaluate the model's ability to handle diverse and multimodal queries, we employ a comprehensive evaluation suite. Our benchmarks include the complete GAIA (Mialon et al., 2024) and XBench-DS (Chen et al., 2025). Additionally, we construct specific test splits from TaskCraft (Shi et al., 2025a) and WebWalkerQA (Wu et al., 2025b). Crucially, the test samples from these datasets are distinct and non-overlapping with the training splits to prevent data leakage. For fair comparison during inference, the underlying LLMs driving the agents include DeepSeek V3.2 (DeepSeek-AI, 2025), Kimi-K2 (Team et al., 2025a), and GPT-5-mini (OpenAI, 2025). We utilize Gemini-3-Flash (Comanici et al., 2025) as the judge model to provide unbiased evaluation of agent trajectories. To validate efficacy, we benchmark `Todo-14B` against a wide spectrum of state-of-the-art systems. Please refer to Table 3 for the detailed list of all baselines compared.

### 5.2. Main Results

**Substantial Performance Enhancement over Baselines.** As presented in Table 3, integrating `TodoEvolve` with the Smolagents framework yields significant performance gains across all evaluated benchmarks. On the comprehensive GAIA benchmark, our approach using GPT-5-Mini achieves an average score of 72.12%, marking a remarkable absolute improvement of 16.37% over the vanilla Smolagents baseline. Furthermore, our method outperforms specialized frameworks operating with the same backbone; for instance, it surpasses Flash-Searcher on GAIA Avg and

*Table 4.* Comprehensive comparison of execution performance across different agent frameworks. The framework achieving the highest accuracy on each benchmark is highlighted in bold.

| Benchmark | Metric | Co-Sight | FlowSearch | Flash-Searcher | AgentOrchestra | OAgents | JoyAgent | OWL | TodoEvolve |
|---|---|---|---|---|---|---|---|---|---|
| **WebWalker-QA** | Accuracy (%) | 16.67 | 30.00 | 60.00 | 46.67 | 33.33 | 63.33 | 53.33 | **70.00** |
| | Avg Cost ($) | 0.0013 | 0.0053 | 0.0134 | 0.0112 | 0.0236 | 0.0028 | 0.0062 | 0.0167 |
| | Avg Time (s) | 190.52 | 94.79 | 164.78 | 137.69 | 150.74 | 212.83 | 127.63 | 216.59 |
| | Avg Step | 2.1 | 4.0 | 5.3 | 6.5 | 7.2 | 4.0 | 3.8 | 7.7 |
| **DeepSearch-QA** | Accuracy (%) | 4.00 | 16.00 | 22.00 | 20.00 | 28.00 | 28.00 | 30.00 | **42.00** |
| | Avg Cost ($) | 0.0025 | 0.0109 | 0.0408 | 0.0263 | 0.0454 | 0.0034 | 0.0191 | 0.0495 |
| | Avg Time (s) | 895.88 | 351.76 | 522.36 | 437.06 | 519.91 | 548.70 | 428.63 | 875.26 |
| | Avg Step | 2.8 | 5.5 | 10.0 | 9.9 | 10.8 | 4.0 | 6.9 | 11.7 |
| **GAIA-level2 Text-only** | Accuracy (%) | 17.14 | 25.71 | 25.71 | 14.29 | 15.71 | 30.00 | 24.29 | **57.14** |
| | Avg Cost ($) | 0.0018 | 0.0069 | 0.0255 | 0.0149 | 0.0317 | 0.0027 | 0.0130 | 0.0282 |
| | Avg Time (s) | 250.23 | 159.14 | 305.67 | 222.75 | 292.12 | 304.38 | 299.78 | 323.65 |
| | Avg Step | 2.6 | 4.6 | 8.0 | 7.7 | 8.7 | 4.1 | 6.2 | 9.1 |

demonstrates superior versatility on domain-specific benchmarks like WebWalkerQA and xBench-DS. These results empirically validate that the autonomous synthesis of task-specific planning architectures offers greater adaptability than static graph-based priors.

**Consistent Gains across Diverse Backbones.** The scalability of `TodoEvolve` is evidenced by its consistent improvements across diverse execution backbones, including GPT-5-Mini, DeepSeek V3.2 and Kimi K2. Notably, when equipped with the DeepSeek V3.2, our framework achieves a GAIA average of 70.91%, significantly outperforming the Flash-Searcher implementation using the same model by over 10 percentage points. This consistency suggests that the meta-planner acquires transferable architectural reasoning capabilities that function independently of the execution model's internal knowledge, effectively acting as a general-purpose performance booster for agentic systems.

**Complex Reasoning with Open-Source Frameworks.** The advantages of `TodoEvolve` are particularly pronounced in high-complexity scenarios requiring long-horizon reasoning. On GAIA Level 3, the most challenging subset, our framework driven by DeepSeek V3.2 attains a success rate of 53.85%. This performance not only surpasses the standard Agent KB using the more powerful GPT-4.1 but also matches the performance of Agent KB with pass@3 voting. This finding highlights a critical insight: with optimal dynamic planning topology, cost-effective open-weights models can rival or exceed the capabilities of resource-intensive proprietary models in complex problem-solving.

### 5.3. Framework Analysis

To dissect the efficacy of our `TodoEvolve`, we conduct a multi-faceted analysis across three critical dimensions: task-dependent structural specialization, execution efficiency, and the individual contribution of our training components.

**Structural Specialization.** We first investigate the performance variability of fixed planning architectures across di-

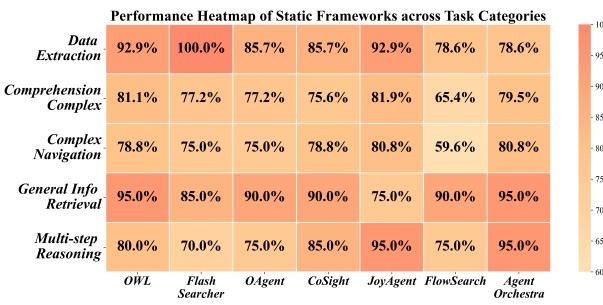

*Figure 2.* Task-Dependent Performance Variability.

verse task typologies, leveraging the GPT-5-mini (OpenAI, 2025) to evaluate a multi-category benchmark extracted from TaskCraft (Shi et al., 2025a). As visualized in Figure 2, distinct planning priors exhibit strong inductive biases suitable for specific domains but lack universality. For instance, centralized systems trade data-handling capacity for reasoning depth, whereas DAG topologies prioritize extraction speed over logical coherence. This heterogeneity highlights a critical limitation that rigid topologies cannot optimally address the structural diversity of open-ended queries. This empirical evidence validates the core premise of `TodoEvolve`: by dynamically synthesizing architectures that integrate the complementary strengths of diverse planning paradigms, our meta-planner achieves cross-domain robustness that no single static framework can match.

**Inference Efficiency.** Beyond task adaptability, we evaluate whether the performance gains of `TodoEvolve` come at the expense of excessive computational overhead. Table 4 details the execution metrics on three benchmarks using the Kimi-K2 (Team et al., 2025a) backbone. `TodoEvolve` consistently achieves dominant accuracy, surpassing the best static baseline by substantial margins (e.g., +10.0% on WebWalker-QA, +14.0% on DeepSearch-QA). Crucially, this performance does not incur a proportional spike in resource consumption, `TodoEvolve` demonstrates superior Pareto optimality: it maintains comparable costs and latency to sophisticated baselines while delivering signifi-

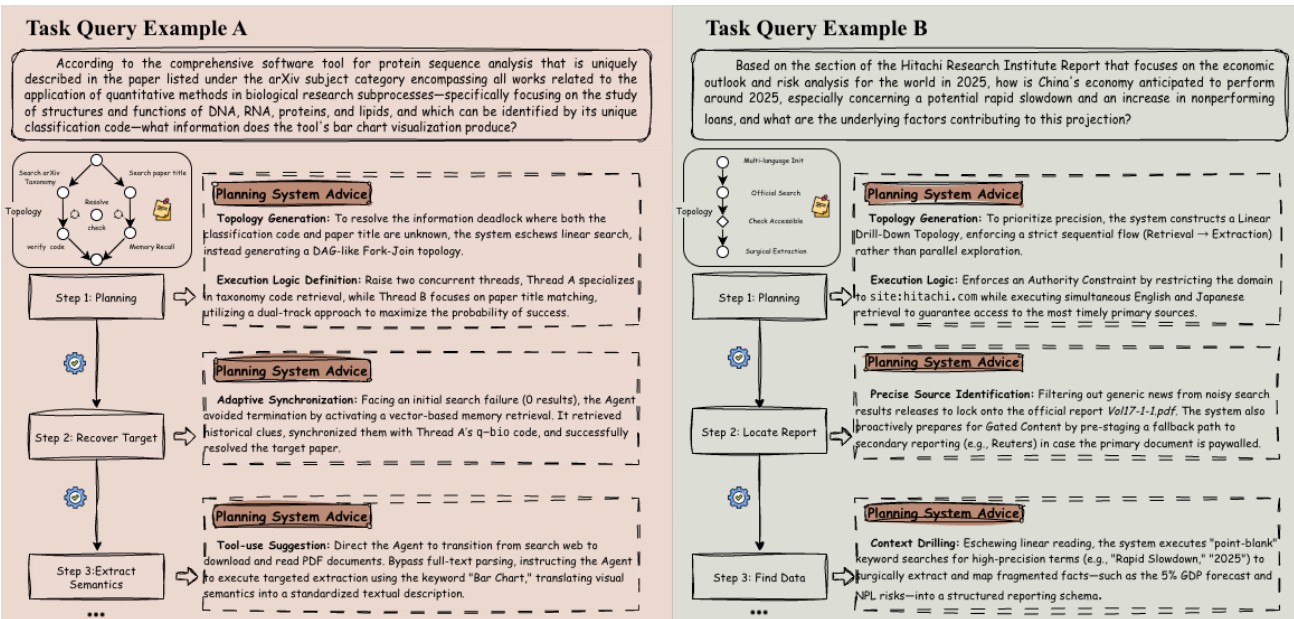

*Figure 3.* Evolved planning architectures in real-world instantiation. The system provides adaptive, state-aware structural scaffolding that spans from macro-topology initialization to granular adaptation and navigation during the execution stage, effectively steering the agent toward robust and resilient inference.

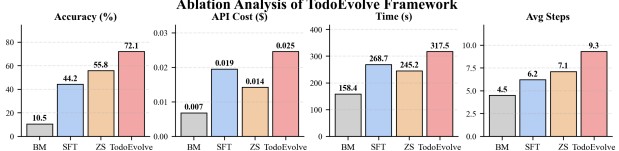

*Figure 4.* Ablation Analysis on GAIA Level 2. We compare the following variants, BS (Base Model), SFT (SFT-Only), ZS (Zero-Shot) and `TodoEvolve`.

cantly higher success rates. This indicates that the meta-planner effectively minimizes cognitive impedance, avoiding the redundant loops of inefficient planners and the premature failures of overly simple ones.

**Ablation Study.** To dissect the efficacy of our training components, we conduct an ablation study on the GAIA Level 2 validation set, comparing four configurations: (1) Base Model, utilizing the unaligned Qwen3-14B to generate planning systems; (2) SFT-Only, fine-tuned on verified planning trajectories; (3) Zero-Shot, which incorporates our IGPO but performs inference without few-shot examples; and (4) `TodoEvolve`, the complete framework employing both training stages and reference-augmented inference. As illustrated in Figure 4, the Base Model fails to synthesize executable plans due to a lack of syntactic grounding, a capability established by SFT-Only. Notably, the Zero-Shot setting improves accuracy to 55.8% while reducing API costs relative to SFT-Only, confirming that IGPO optimizes execution efficiency. Finally, `TodoEvolve` achieves a peak accuracy of 72.1%; the increased steps and cost reflect the planner's enhanced capability to persist through and resolve complex, long-horizon tasks that simpler variants abandon.

## 5.4. Case Study

To illustrate how `TodoEvolve` supports complex reasoning in practice, we present a qualitative analysis of planning systems synthesized during real execution, as shown in Figure 3. Unlike static, "one-size-fits-all" scaffolds that impose a predetermined control flow across all tasks, `TodoEvolve` adapts the planning architecture according to the evolving task state and the structural demands of each query. The resulting plan is therefore not merely a decomposition of the original question, but a task-specific control scaffold that governs evidence collection, uncertainty handling, and execution trajectory revision.

The two examples in Figure 3 illustrate this adaptive behavior from complementary perspectives. In *Task A*, where progress is blocked by distributed and mutually dependent information needs, the meta-planner instantiates a high-breadth Fork-Join topology to explore multiple evidence branches in parallel and then consolidate their outputs. This design helps break information deadlocks and reduces unnecessary sequential waiting. In contrast, for the high-precision objective in *Task B*, the planner enforces a more constrained linear structure, which suppresses irrelevant branches and prunes search-space noise. Moreover, the synthesized planner exhibits predictive resilience by anticipating possible access barriers, such as paywalled reports, and proactively preparing fallback paths to secondary sources. Together, these behaviors suggest that the generated plan serves as a state-aware anchor during execution, mitigating reasoning drift and transforming passive generation into more active, strategic problem solving.

# 6. Conclusion

Traditional agentic planning relies on "one-size-fits-all" workflows, often proving rigid and suboptimal for diverse task demands. This paper aims to transform planning from manual engineering into an autonomous synthesis process, making architectural design as adaptive as the underlying model's reasoning. To this end, we introduce `TodoEvolve`, a meta-planning paradigm that navigates a unified design space, `PlanFactory`, to dynamically configure task-specific topologies and strategies via IGPO. Our extensive evaluations across diverse benchmarks demonstrate that `TodoEvolve` outperforms static baselines, achieving Pareto optimality between success rates and computational efficiency. By bridging the gap between internal reasoning and external architectural scaffolding, `TodoEvolve` provides a blueprint for self-evolving agents capable of mastering open-ended, long-horizon complexities.

## Acknowledgements

The work was in part supported by Scientific Research Innovation Capability Support Project for Young Faculty (U40) of the Ministry of Education of China, SRICSPYF-ZY2025019.

## Impact Statement

This work marks a pivotal shift in agentic AI development: moving from manually engineering static planning workflows to learning how to architect systems autonomously. By demonstrating that a meta-planner can synthesize verifiable and efficient control flows, `TodoEvolve` reduces the reliance on human domain expertise and brittle heuristics. This has profound implications for the democratization of advanced AI agents; our results show that efficient, open-source models equipped with optimal planning structures can rival or even surpass the performance of state-of-the-art models. Consequently, this work contributes to more sustainable AI deployment by explicitly optimizing for planning parsimony, thereby minimizing unnecessary token consumption and latency in real-world applications. Nevertheless, our abstraction does not fully cover all agent orchestration paradigms, particularly highly stateful or memory-centric long-horizon agents that require richer persistent state management. Ultimately, we provide a blueprint for self-evolving agent systems capable of adapting to the open-ended complexity of the real world.

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

## A. `PlanFactory` Details

We detail the established planning system in `PlanFactory` as follows:

**Overview of Representative Frameworks.** To clarify the planning styles covered by `PlanFactory`, we first summarize the representative systems decomposed in our design space. Flash-Searcher (Qin et al., 2025) and OAgents (Zhu et al., 2025) are based on explicit task dependencies: the plan is decomposed into linked subtasks, and execution follows prerequisite relations. OWL (Hu et al., 2025b) and AgentOrchestra (Zhang et al., 2025b) adopt a more hierarchical style, where a high-level planner decomposes the objective and dispatches work to lower-level workers. JoyAgent (Liu et al., 2025) is closer to an ensemble setting, where multiple expert teams work in parallel and their outputs are combined through voting, criticism, or synthesis. FlowSearch (Hu et al., 2026b) centers on a dynamically evolving knowledge structure that is expanded and refined during execution. Co-Sight (Zhang et al., 2025a) is verification-oriented, emphasizing cross-checking, inconsistency detection, and conflict resolution before advancing the plan. In summary, these systems share a common planning abstraction but differ in how they organize, revise, and execute plans.

- **Co-Sight**

  Co-Sight establishes a cross-check net topology, specifically engineered to resolve epistemic discrepancies through mutual verification. The system is initialized via an inconsistency trigger, where the planning process is activated only upon detecting conflicting information or divergent perspectives among internal modules. Navigation is executed through conflict resolution, utilizing trustworthy reasoning with structured facts to systematically eliminate cognitive biases across the agent collective. For its adaptation mechanism, the framework employs meta-verification, conducting high-level assessments of the underlying verification logic to ensure the integrity of the process of building consensus.

- **AgentOrchestra**

  AgentOrchestra adheres to an orchestration hierarchy topology, establishing a structured command chain for multi-agent coordination. The system initiates through role definition, where functional identities are assigned to activate the environment. During this phase, a planning agent leverages its global perspective to decompose complex objectives into manageable sub-tasks. Navigation is facilitated via centralized routing, with the planning agent dispatching specific instructions to specialized sub-agents based on their designated roles. The framework's adaptation is driven by environment feedback, where the system dynamically re-calibrates the plan by synthesizing execution data, aggregating feedback loops, and monitoring cumulative progress toward the final objective.

- **OAgents**

  OAgents employs a modular graph topology, representing the global objective as a web of decoupled yet interdependent modules. The framework initiates via SOP configuration, where the agent decomposes the primary task into sub-tasks interconnected by edges that define prerequisite dependencies. Navigation is driven by dynamic programming, which, at each discrete step, identifies and dispatches the set of candidate nodes whose dependencies have been fully satisfied. The system's adaptation mechanism relies on critic-loop feedback for periodic refinement: every $N$ steps, intermediate results are cross-referenced against global constraints to verify alignment with the objective, triggering a re-sequencing of sub-tasks based on novel observations. Furthermore, trajectories from prior execution attempts are distilled into heuristic guidance and integrated into the planning module as soft constraints or behavioral preferences, dynamically biasing sub-task selection toward proven success paths.

- **JoyAgent**

  JoyAgent utilizes a collective hierarchy topology, structuring its multi-agent system to balance global oversight with local flexibility. the system is initialized through hybrid planning, which implements a supervisor agent based on a plan-and-execute framework to maintain global coherence while concurrently deploying multiple single agents utilizing react to ensure step-level responsiveness. navigation is governed by joint deliberation, where outputs from the diverse agent pool are aggregated and processed through consensus voting to determine the optimal execution path. the framework's adaptation is achieved through the intrinsic react loops of the individual agents, allowing for real-time adjustments based on localized feedback without compromising the overarching trajectory.

- **Flash-Searcher**

  Upon receiving a request, Flash-Searcher decomposes the task into a parallel Directed Acyclic Graph (DAG), where nodes denote granular sub-tasks and edges represent their dependencies. The system instantiates this structure through

dependency parsing, mapping out the prerequisite constraints to initialize the graph's nodes and edges. Navigation is governed by aggressive parallelization. A node is dispatched to a concurrent execution pool as soon as its predecessors are satisfied or when partial execution results provide sufficient auxiliary validation. To maintain system agility, the framework performs workflow pruning at defined step intervals, where it summarizes progress to excise resolved nodes and re-evaluates the dependencies of pending tasks, dynamically injecting new decomposition branches if environmental contingencies arise.

- **FlowSearch**

  FlowSearch conceptualizes task resolution through a thought graph topology, representing reasoning as an evolving network of cognitive states. The framework employs flow construction for incremental instantiation; starting from the root task, a knowledge flow planner iteratively evaluates whether active nodes require further decomposition or supplemental context. This process generates descendant nodes that encapsulate sub-problems, intermediate reasoning steps, and evidentiary grounding while establishing dependency edges to preserve logical consistency and structural integrity. Navigation is managed by a knowledge collector, which identifies and dispatches nodes with the highest execution readiness based on satisfied dependencies. The system's adaptation is realized through dynamic expansion via a knowledge refiner, which leverages newly acquired insights to perform structural transformations on the flow. By synthesizing current knowledge contexts with execution states, the refiner dynamically executes atomic operations including the addition, deletion, or modification of nodes and edges to optimize the graph's trajectory toward the goal.

- **OWL**

  OWL adopts a dual hierarchy topology that formally segregates the strategic management layer from the tactical execution layer. Upon task arrival, the system undergoes planner decomposition, where a high-level planner analyzes task complexity against the latent capabilities of available worker nodes to instantiate a structured task list. Navigation is facilitated via dynamic dispatch, managed by a coordinator that evaluates real-time agent profiles to map specific sub-tasks to the most suitable worker nodes. The framework's adaptation logic is driven by manager intervention triggered by decentralized failure detection: individual workers autonomously monitor their execution status, broadcasting failure signals to a dedicated task channel upon impasse. This channel acts as an observation primitive, prompting the planner to perform reactive re-planning and inject revised sub-tasks based on the contextual feedback from the failed execution.

## B. Datasets

The five datasets used in this study are described as follows: (1) **GAIA** (Mialon et al., 2024) consists of 165 tasks, categorized into 53 Level-1, 86 Level-2, and 26 Level-3 problems. (2) **WebWalkerQA** (Wu et al., 2025b) evaluates an agent's capability in handling complex, multi-turn web interactions. It comprises 680 real-world queries across four domains and spans over 1373 webpages. We sample a subset of 170 queries for evaluation. (3) **xBench-DeepSearch (xBench-DS)** (Chen et al., 2025) contains 100 tasks assessing agentic planning, tool use, and reasoning. (4) **TaskCraft** (Shi et al., 2025a) is a synthetic benchmark generated via an autonomous data pipeline, we collect 300 queries as a valid subset. (5) **DeepSearchQA** (Google DeepMind, 2025) targets the long-horizon research capabilities of agents, we collect 50 queries as a valid subset.

## C. Data Construction Cost

We report the budget of the training-set construction pipeline to clarify the relationship between distillation cost and downstream improvement. Data collection is performed over 2,500 distinct queries, with two independent evolutionary sampling runs per query, yielding 5,000 initial candidate trajectories with full execution traces and logged metrics. From this candidate pool, we retain 3,360 verifiable trajectories for SFT and construct 2,000 high-quality IGPO preference pairs.

*Table 5.* Data construction cost computed from logged token usage.

| Model Role | Total Input Cost ($) | Total Output Cost ($) |
|---|---|---|
| Gemini-3-Flash Planner | 396.97 | 134.89 |
| DeepSeek V3.2 Executor | 679.45 | 478.95 |
| Gemini-3-Flash Judge | 83.71 | 69.58 |

This cost is incurred offline during data construction. At test time, TodoEvolve performs one-shot planner generation and does not require evolutionary search over multiple candidate planners.

# D. Framework and Evaluation Details

### D.1. End-to-End Pipeline

`TodoEvolve` does not directly produce the final answer. Instead, the meta-planner first generates a task-specific planning system, which is then executed by the downstream agent within `PlanFactory`. This separation distinguishes planning-system generation from answer generation.

For each task instance, the meta-planner receives a task-specific planning prompt consisting of: (1) the user query; (2) the role-defining system prompt; (3) the abstract `BasePlanning` interface; (4) generation objectives and formatting constraints; (5) framework-level runtime specifications; (6) three randomly sampled planning examples from the decomposed planning frameworks; and (7) descriptions of callable tools. The output of the meta-planner is not an answer, but a generated Python planning module together with a YAML configuration.

At execution time, the downstream agent receives: (1) the original question; (2) the generated planning module; (3) the generated YAML configuration; and (4) the callable tools and their descriptions. The agent then executes under this planner/configuration within PlanFactory and produces the final answer.

### D.2. Planning Function Interface

PlanFactory formalizes the planning lifecycle through four dimensions: Topology, Initialization, Adaptation, and Navigation, which respectively define how a planner is structured, initialized, revised, and translated into executable directives.

> **Planning Function Interface**
>
> ```
> topology(self, task:  str) -> PlanningStep
> initialize(self, task:  str) -> PlanningStep
> adaptation(self, task:  str, step:  int, write_memory:  Callable) -> SummaryStep
> navigation(self, task:  str, step:  int, write_memory:  Callable) -> ActionStep
> ```

The runtime is recorded through three step objects: `PlanningStep` for topology and initialization outputs, `SummaryStep` for progress summaries and plan revisions, and `ActionStep` for executable navigation steps. This interface makes heterogeneous planners interchangeable within the same runtime, allowing generated planners to update state through `SummaryStep` and issue execution guidance through `ActionStep`.

### D.3. Evaluation Protocol

For tasks with labeled answers, we use an LLM-as-a-Judge protocol to evaluate whether the predicted answer is equivalent to the ground-truth answer. The judge is instructed to output a structured JSON object with a rationale and a binary judgement.

> **LLM-as-a-Judge Prompt**
>
> ```
> Please determine if the predicted answer is equivalent to the labeled answer.
> Question:  {question}
> Labeled Answer:  {gt_answer}
> Predicted Answer:  {pred_answer}
> Are these answers equivalent?  The output should be in the following JSON format:
> {
>   "rationale":  "your rationale for the judgement, as a text",
>   "judgement":  "your judgement result, can only be 'correct' or 'incorrect'"
> }
> ```

# E. Case Study

To provide a concrete and intuitive understanding of the planning architectures synthesized by `TodoEvolve`, we visualize three representative systems generated for distinct query types, as shown in Figures 5 to 7. These examples demonstrate how our meta-planner moves beyond static templates by dynamically tailoring control flows from linear sequential logic to complex parallel graph structures, matching the specific cognitive impedance and dependency requirements of each task.

By autonomously configuring topology initialization, execution navigation, and adaptation triggers, `TodoEvolve` ensures robust performance across varying levels of problem complexity.

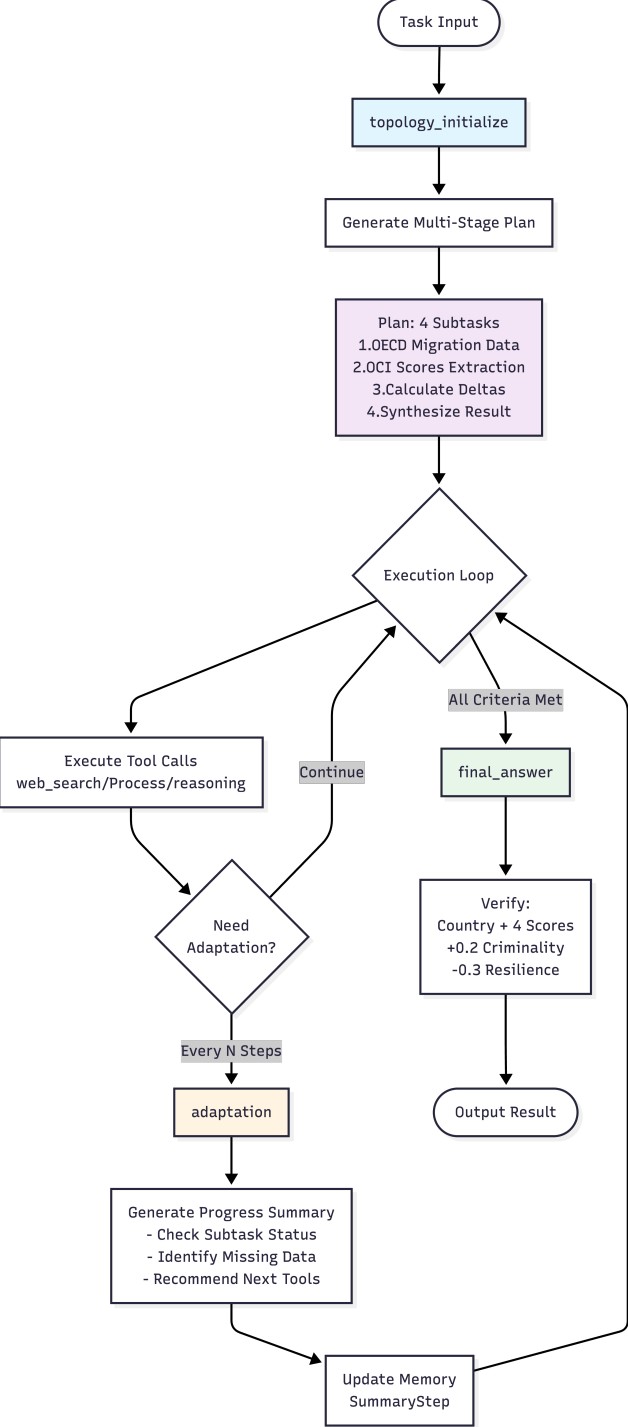

*Figure 5.* **Linear Sequential Planning for Multi-Criteria Filtering.** For a query requiring strict multi-stage filtering and calculation (identifying countries based on migration thresholds followed by crime index analysis), `TodoEvolve` instantiates a linear execution topology. The system prioritizes a sequential "fetch-and-filter" pipeline to manage data dependencies, incorporating a periodic adaptation trigger to validate intermediate retrieval results before proceeding to the final synthesis and verification stage. This structure minimizes branching overhead for tasks where step-wise logical progression is paramount.

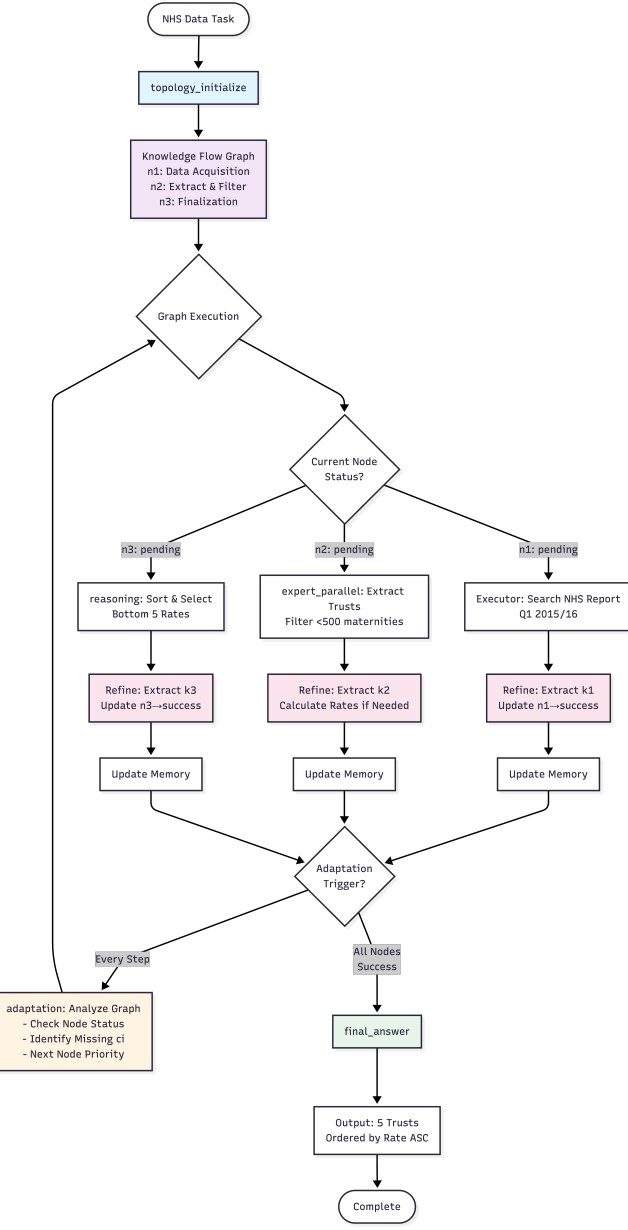

*Figure 6.* **State-Aware Graph Topology for Structured Data Extraction.** Addressing a structured retrieval task involving sorting and ranking constraints, the meta-planner constructs a Knowledge Flow Graph. This topology decomposes the problem into granular nodes (acquisition, filtering, and finalization). The navigation strategy employs a state-aware routing mechanism that dynamically selects between parallel extraction or sequential reasoning based on the current node status ("pending" vs. "success"), allowing the system to efficiently prune the search space while adhering to numerical constraints.

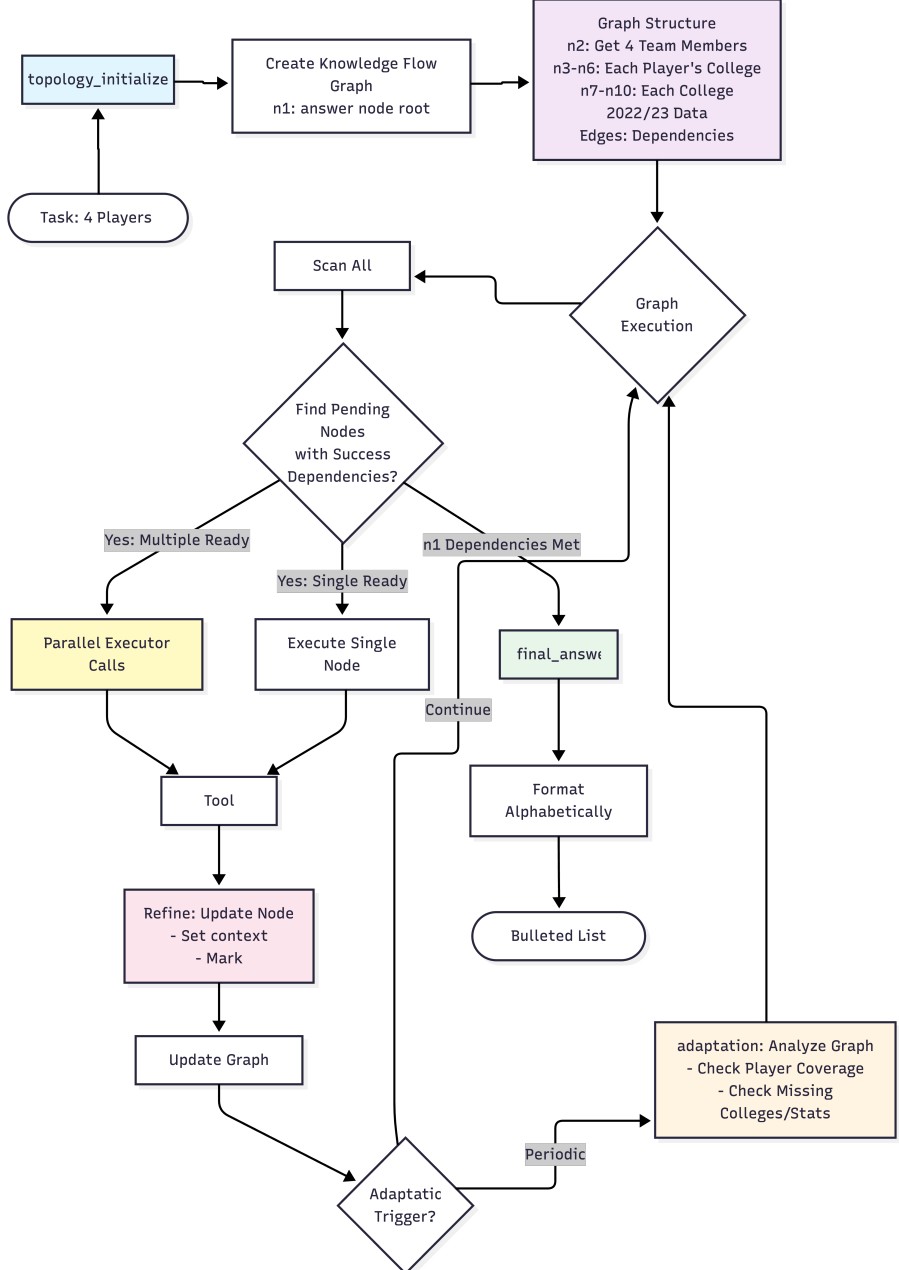

*Figure 7.* **High-Breadth Parallel Planning for Complex Entity Resolution.** Faced with a complex entity resolution task requiring the retrieval of nested attributes for multiple subjects simultaneously, `TodoEvolve` evolves a highly parallelized graph architecture. The system identifies independent sub-goals (e.g., retrieving data for different players concurrently) and activates a "Parallel Executor" module to minimize latency. The adaptation layer monitors the synchronization of these concurrent streams, ensuring that the graph topology is only updated and merged when specific dependency conditions are met.

