# OpenReview forum: "TodoEvolve: Learning to Architect Agent Planning Systems"
_ICML.cc/2026/Conference — ICML 2026 regular_

### Official Review · Reviewer_yjoi · 2026-03-05

**Soundness:** 3
**Presentation:** 3
**Significance:** 3
**Originality:** 3
**Overall Recommendation:** 5
**Confidence:** 4

**Summary:**

This paper investigates that existing agent systems rely too heavily on a single planning structure. The authors first introduce PlanFactory, which comprises four components: topology, initialization, adaptation, and navigation. They then propose TodoEvolve. Based on this, they train Todo-14B (a meta planner across diverse scenarios and backbones) via IGPO. Finally, the effectiveness of the proposed method is validated on four challenging benchmarks.

**Compliance With Llm Reviewing Policy:**

Affirmed.

**Final Justification:**

Thanks to the authors for their response. The rebuttal has addressed my concerns, so I have raised my score.

**Key Questions For Authors:**

(1) For frontier commercial large models, it is impossible to rely too heavily on distillation from stronger models. In other words, how can the meta planner capabilities of these commercial large models be further improved when there is no teacher model significantly stronger than the current model? Could the authors provide some insights regarding this situation?

**Limitations:**

yes

**Strengths And Weaknesses:**

Strengths

(1) This framework can break away from traditional, single planning structures to achieve a unified planning framework. This motivation shows broad and promising prospects.

(2) PlanFactory abstracts four components, including topology, initialization, adaptation, and navigation, presenting a clear and explicit structure.

(3) The authors conduct an analysis of Accuracy, Cost, and Time, which is crucial for practical deployment and usage.

Weaknesses

(1) High quality data construction relies on Gemini-3-Flash as the expert planner and DeepSeek V3.2 as the executor. The capabilities of the student models are constrained by the upper bound of these teacher models' capabilities, which introduces high costs for data scalability. Please report the training set construction costs required when using Gemini-3-Flash and DeepSeek V3.2 as the expert planner, executor, and judge model. This will allow readers to understand the relationship between the distillation budget and performance improvements.

(2) It remains unclear whether using open source large models as teacher models can improve the capabilities of smaller models.

(3) The authors only trained Qwen3-14B. It is unclear whether this method possesses generalizability across different backbones and parameter scales.

---

> ### Author Rebuttal · Authors · 2026-03-31
>
> Dear Reviewer yjoi,
>
> We sincerely appreciate your constructive review! In the following, we present detailed responses to your concerns.
>
> ## Q1. Training-set construction cost
>
> Thank you for raising this practical point. We agree that computational budget report is important for assessing the scalability of our work. The data collection is performed over **2500 distinct queries**, with **two** independent evolutionary sampling runs per query, yielding **5000 initial candidate trajectories** together with full execution traces and logged metrics.
>
> Our pipeline also achieves high data efficiency, from this initial candidate pool, we retain 3,360 verifiable trajectories for SFT (66.8% retention) and construct 2,000 high-quality IGPO preference pairs (80% retention), which shows that the API budget is effectively converted into useful training signals.
>
> As shown in the table below, we present the exact data-collection cost breakdown, which is computed based on recorded token usage:
>
> | Model (Role) | Total Input Cost (\$) | Total Output Cost (\$) |
> | -------- | -------- | -------- |
> | Gemini-3-Flash (Planner)        |  396.97        |   134.89       |
> | DeepSeek V3.2 (Executor)         |   679.45    |   478.95       |
> | Gemini-3-Flash (Judge)     | 83.71   |   69.58   |
>
> ---
>
> ## Q2. Teacher models choices
>
> Thank you for this insightful question. We agree that exploring open-source teacher models is a meaningful direction. In this work, we use a strong closed-source teacher to ensure sufficient **data quality**. In particular, IGPO requires valid preference pairs constructed from successful executions. If the model fails too often, the number of usable preference pairs drops sharply, which directly limits training effectiveness.
>
> But TodoEvolve is **not** tied to a specific teacher family, open-source teachers can be substituted, provided they can generate enough valid training data. The key remaining research question is not framework compatibility, but mapping the exact trade-offs between teacher reasoning strength, data quality, and downstream student improvement.
>
> We believe your question also points to a promising future direction: enabling smaller open-source models to improve their meta-planning ability **through iterative self-improvement**. We will make this discussion in the revised manuscript.
>
> ---
>
> ## Q3. Training backbones generalization
>
> Thank you for this important question. We agree that it is crucial to clarify the scope of our method's generalizability. We distinguish between two dimensions of generalization:
>
> - **Execution-Side Generalization**: whether a single trained meta-planner remains effective across different downstream execution models. Our paper directly supports this: the trained **Qwen3-14B** meta-planner consistently improves performance across **GPT-5-mini, DeepSeek V3.2, and Kimi-K2**.
>
> - **Training-Side Generalization**: whether the TodoEvolve training pipeline itself transfers across different meta-planner backbones and scales.
>
> To address this more directly, we applied our training pipeline to two additional backbones and evaluated the resulting planning systems on GAIA Level-2, using GPT-5-mini as the shared execution backbone:
>
> | Backbone                  | Training Stage | Accuracy (%) |
> |:------------------------- |:-------------- |:------------ |
> | seed-coder-8b                          |    base            |   4.65           |
> | seed-coder-8b             | SFT Only       | 22.09        |
> | seed-coder-8b             | SFT + IGPO     | **48.83**        |
> | Qwen3-14b                          |     base           |      15.12        |
> | Qwen3-14b                 | SFT Only       | 44.18        |
> | **Qwen3-14b(TodoEvolve)** | SFT + IGPO     | **72.09**        |
> | GPT-OSS-20b                          |  base              |    19.77          |
> | GPT-OSS-20b               | SFT Only       | 27.91        |
> | GPT-OSS-20b               | SFT + IGPO     | **61.62**        |
> These results confirm that our training paradigm effectively and universally generalizes to different backbones, providing consistent capability boosts.
>
> ---
>
> ## Q4. Further improvement
>
> Thank you for this forward-looking question. We agree that distillation is unlikely to remain the only path for improving frontier meta-planners. One advantage of our formulation is that **planning quality can be evaluated through downstream execution outcomes**, which makes it naturally compatible with RLVR[1].
>
> In this setting, future direction may come from **agent self-improvement**, for example through self-distillation, hard-example mining or using trial-and-error to find better solutions. We therefore view TodoEvolve as a natural starting point for future LLM planning system optimization with verifiable rewards. We will make this future-work direction more explicit in the revision.
>
> ---
>
> ### References:
> [1] DeepSeek-R1 incentivizes reasoning in LLMs through reinforcement learning
>
> Sincerely,
>
> Authors of Submission 5400

---

> > ### Author Rebuttal · Reviewer_yjoi · 2026-04-03
> >
> > Thanks to the authors for their response. The rebuttal has addressed my concerns, so I have raised my score.

---

> > > ### Author Response · Authors · 2026-04-04
> > >
> > > Dear Reviewer yjoi,
> > >
> > > We thank you for your kind follow-up and for raising your score. We greatly appreciate your recognition that our rebuttal addressed your concerns. Your encouragement is very motivating, and we are glad that the explanations regarding training costs, teacher model choices, backbone generalization, and future directions clarified your questions.
> > >
> > > Thank you again for your thoughtful reading and support!
> > >
> > > Sincerely,
> > >
> > > Authors of Submission 5400

---

### Official Review · Reviewer_ZQdY · 2026-03-13

**Soundness:** 2
**Presentation:** 1
**Significance:** 3
**Originality:** 3
**Overall Recommendation:** 2
**Confidence:** 3

**Summary:**

The paper attempts to unify different agent architectures and task hierarchies into a framework defined by topology, instantiation mechanism, adaption logic and execution directive. It attempts to propose a training formalism IGPO to improve the meta-planner, and compares the performance of the proposed method against several benchmark.

**Compliance With Llm Reviewing Policy:**

Affirmed.

**Key Questions For Authors:**

1. While one of the main contribution of the paper is a unified framework for various Agentic AI Architectures, it does not spend enough time explaining how the various existing architectures (shown in Table 1) align with the proposed framework. Also all of these agents are not well established, so I expect some background on that in section 2.
2. The term “Planning Structure, “meta planner” is vague. To me the structure that the proposed “Plan Factory” is trying to unify is agent architecture (agent system design) or orchestrator design. I would highly recommend authors to restate the paper as an agent-system designer.
3. It would be helpful to have citation of methods and baselines in the Tables. There seems to be inconsistency in the citations. AgentOrchestra is cited as Li et al 2025a in Section 2, but the paper in reference is AOP—Agent oriented planning. What is the correct citation for AgentOrchestra?
4. What is the motivation being “Cognitive Impedance”? The equation is not well explained. Which errors do $N_{fail}$ counts? $S_{stab}$ quantifies execution smoothness — what does execution smoothness mean?
5. Experiment Setup Section 5.1 mentions Qwen3-14B as backbone for our model but the results in the table mention model family GPT-5-mini, Kimi-K2, and Deepseek V3.2. So which model is used where.
6. Why is there a difference in the baseline compared in Table 2 vs 3? It is not clear what multi-agent architecture is used with the framework mentioned in Table 3. Is Table 4  average over multiple runs and Table 3 pass@1 ?

**Limitations:**

yes

**Strengths And Weaknesses:**

Soundness: The paper seems technically sound, but it lacks critical details at this time for me to fully confirm.

Presentation: The paper lacks clarity. See Questions

Significance: The work has the potential to impact future research but the paper is too vague in its current state to make sense.

Originality: Not a concern at this stage.


**Strengths**: Paper shed a different light at multi-agent system orchestrations. It has potential to impact future research.

**Weakness**: Lacks clarity

---

> ### Author Rebuttal · Authors · 2026-03-31
>
> Dear Reviewer ZQdY,
>
> Thank you for your careful reading and insightful feedback.
>
> ## Q1. Framework Mapping Clarification
>
> Thank you for this important point. We agree that Table 1 should be an explicit factorization of representative planning systems under our abstraction $P=(G, I_{\text{init}}, F_{\text{adapt}}, N_{\text{nav}})$, corresponding to **topology, initialization, adaptation, and navigation**. The key point is that these frameworks can be described by the same four components.
>
> For example:
>
> - **AgentOrchestra**[1], $G$ is a hierarchical task list, $I_{\text{init}}$ decomposes the objective into prioritized subtasks, $F_{\text{adapt}}$ updates subtask status from execution progress, and $N_{\text{nav}}$ dispatches the next unfinished subtask/worker.
>
> - **FlowSearch**[2], $G$ is a dependency graph over knowledge nodes, $I_{\text{init}}$ expands the graph from the task, $F_{\text{adapt}}$ updates node status/context after observations, and $N_{\text{nav}}$ selects ready nodes based on dependencies.
>
> These systems are re-implemented in PlanFactory as structural references and baselines. We agree the mapping is implicit, so we will clarify our PlanFactory backgrounds and explanations in the revision.
>
> ---
>
> ## Q2. Terminology: Planning Structure vs Meta-Planner
>
> Thank you for this highly constructive feedback. We agree that current claim might unintentionally suggest that we are simply  another static agent workflow or merely collecting superficial agent trajectories. We clarify these two concepts below:
>
> - **Planning Structure**: The high-level execution blueprint that organizes how the agent plans, updates, and acts during a task. Prior frameworks typically rely on fixed, manually designed structures (e.g., linear flows, DAGs, or static multi-agent workflows).
>
> - **Meta Planner**: The trained generative model (Todo-14B) that synthesizes such query-specific planning systems dynamically, which enhances the downstream agent's task comprehension and execution ability.
>
> We agree that our contribution is better framed as agent planning architecture design, with two properties: Our work focuses on the agent's **planning** layer, demonstrating this capability can be **learned** through training instead of manual engineering.
>
>
> ---
>
> ## Q3. Cognitive Impedance Motivation
>
> Thank you for raising this point. As this overlaps with Reviewer f6H5’s Q4, we kindly refer you to that response due to space limits. It details the motivation for Cognitive Impedance, the operational variables($N_{fail}$, $S_{stab}$, and $\frac{C_{plan}}{C_{exec}}$), and the ablation results.
>
>
>
> ---
>
> ## Q4. Backbone and Evaluation Models
>
> Thank you for highlighting this ambiguity. A core feature of our framework is the strict decoupling of the meta-planning layer from the downstream execution agent.
>
> In Table 3, the “Model Family” column refers only to the downstream execution model used at evaluation time, whereas the trained meta-planner is consistently based on Qwen3-14B across all experiments.
>
> The roles of the models that appeared in our work are as follows:
>
> | Model | Role |
> | ----- | ---- |
> | Qwen3-14B       |  Meta-planner backbone (Todo-14B)     |
> | Gemini-3-Flash      |  Teacher during data construction; Judge during evaluation    |
> | DeepSeek V3.2      | Executor during data construction; one evaluation-time execution backbone.     |
> | GPT-5-mini/Kimi-K2      | Evaluation-time execution backbones        |
>
>
> ---
>
> ## Q5. Evaluation Protocols
>
> Thank you for this question. We believe you may be referring mainly to Tables 3 and 4, since Table 2 reports dataset statistics. Their purposes are **different**:
>
> - Table 3 is our **main results table**, comparing TodoEvolve against a broad set of recent agent planning systems across benchmarks and execution backbones.
>
> - Table 4 is a **controlled comparison**, against the 7 representative planning systems decomposed in our PlanFactory, to show that TodoEvolve outperforms the static planning templates used as structural references.
>
> Both Table 3 and Table 4 report **single-run pass@1** accuracy unless otherwise noted. The “Avg” metrics in Table 4 (Cost / Time / Step) are not averages over repeated runs. They are averages over all test queries in a single benchmark run, showing that the gains do not come from inflated overhead.
>
> We will clarify these experiment settings more explicitly in the revision. We sincerely thank the reviewer again for the constructive feedback on the presentation and clarity of our paper.
>
> ---
>
> ## Q6. Citation Consistency
>
> Thank you for catching this oversight, you are correct that "Li et al. 2025a" refers to AOP rather than AgentOrchestra. We will correct this citation and standardize method references across all tables.
>
> ---
>
> ### References:
> [1] AgentOrchestra: Orchestrating Multi-Agent Intelligence with the Tool-Environment-Agent(TEA) Protocol
> [2] FlowSearch: Advancing deep research with dynamic structured knowledge flow
>
>
> Sincerely,
>
> Authors of Submission 5400

---

> > ### Author Rebuttal · Reviewer_ZQdY · 2026-04-01
> >
> > Thank you for addressing the questions. To justify the paper’s claim as a unified framework for diverse Agentic AI architectures, a formal and comprehensive mapping of the framework is essential and should be clearly presented and reviewed. Addressing this issue would require a substantial rewrite of the paper and cannot be adequately handled within the rebuttal. Therefore, I will maintain my original score.

---

> > > ### Author Response · Authors · 2026-04-02
> > >
> > > Dear Reviewer ZQdY,
> > >
> > > Thank you for the follow-up. We believe the remaining concern is primarily about the **scope of our claim**, and we would like to make this boundary fully explicit. The technical objective we are pursuing is **not** a universal abstraction of all **agent-system** designs, but a unified framework for agent **planning systems**.
> > >
> > > In our earlier rebuttal, we already tried to state this in two places, although we may not have emphasized it enough.
> > >
> > > - First, in **Q1 Framework Mapping Clarification**, we wrote that “**Table 1 should be an explicit factorization of representative planning systems under our abstraction**.”
> > > - Second, in **Q2 Terminology**, we stated that “**our contribution is better framed as agent planning architecture design**.” In other words, the object unified by PlanFactory is the **planning-system layer**, rather than arbitrary end-to-end agent-system architectures.
> > >
> > > This scope is also consistent with the paper itself. The title is **“Learning to Architect Agent Planning Systems”**, and the main text formulates PlanFactory under a bi-level abstraction in which the Planning System is **separated** from the downstream Agent System. Within this abstraction, PlanFactory is defined as a **unified planning** codebase built around four planning components: **Topology**, which defines the structural organization of the plan; **Initialization**, which specifies how the plan is instantiated from the task; **Adaptation**, which governs when and how the plan is revised during execution; and **Navigation**, which determines how the plan produces executable directives for the acting agent.
> > >
> > > This interpretation is also consistent with how the other reviewers described the contribution.
> > >
> > > - Reviewer g2ND explicitly wrote that the paper provides a “**good abstraction of the planning mechanisms**” and described TodoEvolve as “**a meta-planning paradigm that learns to instantiate and control planning structures on the fly**.”
> > > - Reviewer f6H5 likewise summarized the work as introducing “**a sufficient generic formulation for describing agentic planning architecture**” via $P=\langle G, I_{\text{init}}, F_{\text{adapt}}, N_{\text{nav}}\rangle$
> > > - Reviewer yjoi also referred to the work as achieving “**a unified planning framework**” and specifically highlighted the four planning components as “**a clear and explicit structure**.”
> > >
> > > Based on the discussion above, we therefore believe that our framework is expressive enough to describe and subsume the mainstream **planning systems** currently used in agent practice. This is not a post-hoc narrowing of the claim, nor a compromise made only for the rebuttal, rather, it reflects the technical object of the paper from the beginning: our work has always focused on the **decomposition, re-composition, and learning of planning systems**, rather than on a universal abstraction of end-to-end agent-system designs. If the reviewer has a specific planning system in mind that appears not to fit this abstraction, we would be grateful for that example and would be happy to explain either how it maps into our framework or where the true boundary should be drawn.
> > >
> > > We sincerely thank the reviewer for the continued engagement in this discussion. We greatly value the opportunity to clarify this core conceptual point, and would be happy to continue the exchange further if useful.
> > >
> > > Faithfully,
> > >
> > > Authors of Submission 5400

---

### Official Review · Reviewer_f6H5 · 2026-03-13

**Soundness:** 3
**Presentation:** 2
**Significance:** 3
**Originality:** 3
**Overall Recommendation:** 5
**Confidence:** 4

**Summary:**

This paper proposes TodoEvolve, a meta-planning framework that automatically synthesizes planning architectures for LLM agents. Instead of relying on a fixed planner (e.g., ToT, DAG planners, hierarchical systems), the method generates task-specific planning modules along four dimensions: topology, initialization, adaptation, and navigation. The system is implemented via a unified framework called PlanFactory, and a 14B meta-planner model (Todo-14B) is trained using a two-stage pipeline consisting of supervised fine-tuning followed by Impedance-Guided Preference Optimization (IGPO). Experiments across several agent benchmarks demonstrate improvements over static planning frameworks.

**Compliance With Llm Reviewing Policy:**

Affirmed.

**Final Justification:**

The new experiments show the effective of cognitive impedance although its design is empirical.

**Key Questions For Authors:**

* Generalization: How well does the meta-planner generalize to entirely new domains or tools not seen during PlanFactory data construction?

* Extendability: How difficulty would it be extend the PlanFactory with new domains?

* To what extent are the improvements due to training the meta-planner, versus simply searching many planning candidates during data generation? Did you compare against a test-time search baseline that samples multiple planners and selects the best?

**Limitations:**

* Limited planner search space: Although the paper frames the problem as learning to synthesize planning architectures, the generated planners appear constrained by the PlanFactory design space, which defines four components (topology, initialization, adaptation, navigation).

* Small training dataset for a complex task: The training dataset used for meta-planner training appears relatively small: ~3,360 SFT samples and ~2,000 preference pairs for IGPO.

* Heuristic definition of cognitive impedance

**Strengths And Weaknesses:**

**Strength:**

* The technical framing of the work is generally sound and well motivated. The paper introduces a sufficient generic formulation for describing agentic planning architecture, $P= \langle G, I_{init}​,F_{adapt}​,N_{nav}​ \rangle$ corresponding to topology, initialization, adaptation, and navigation components.

* The paper also demonstrates sufficient engineering complexity to curate the dataset and post-training LLMs as meta-task planner generator. These engineering effort includes bootstrap-and-filtering for data curation. The introduction of The IGPO objective. A general composite efficiency metric called cognitive impedance, incorporating cost, failures, stability, and planning overhead.

* The empirical evaluation includes multiple benchmarks (GAIA, xBench-DS, WebWalkerQA, TaskCraft) and several LLM backbones. The reported improvements appear consistent across backbones, suggesting that the meta-planner provides architecture-level improvements rather than model-specific gains.

The ablation studies (Base Model vs SFT vs IGPO vs full method) help isolate the contribution of each component and provide evidence that the preference optimization stage improves efficiency.

**Weakness**

* PlanFactory involves many planning domains, each may have their own verification and sandbox environment for execution. This raises the concern about extendability (adding more planning domains) and engineering barrier for executing all the plans across different domains.

* Definition of Cognitive Impedance: The impedance metric combines multiple factors via an exponential formulation, but the paper does not justify the coefficients or provide sensitivity analysis.

**Presentation**

* The overall structure is clear.

* The use of some terms are not very clearly defined. For example, when topology, initialization, adaptation, and navigation components are first mentioned in the introduction, they are not clearly defined. Also includes terms like "architectural reasoning". This makes the paper sometimes hard to follow.

* Also, the use of certain techniques are not very motivated when they are first mentioned (e.g., evolutionary sampling in line 192).

---

> ### Author Rebuttal · Authors · 2026-03-31
>
> Dear Reviewer f6H5,
>
> Thank you for your thoughtful feedback.
>
> ---
>
> ## Q1. Domain / tool generalization
>
> Thank you for this concern. Our current setup already partially provides preliminary evidence of transfer to unseen tools and domains:
>
> - **Tool Generalization**: Train data mainly emphasizes text-based retrieval and reasoning, whereas GAIA requires code execution and multimodal tool calls, TodoEvolve still yields strong gains under this tool shift.
>
> - **Domain Generalization**: xBench-DS is fully excluded from training and focuses on Chinese daily-life scenarios, differing substantially from the training benchmarks, consistent gains there support held-out domain transfer.
>
> Thus, we view these results as preliminary rather than definitive evidence of broad domain/tool generalization. Stronger claims in this direction will require more targeted experiments and further engineering, and we will clarify this boundary in the revision.
>
> ---
>
> ## Q2. PlanFactory
>
> Thank you for this point. PlanFactory standardizes planning systems through four dimensions—topology, initialization, adaptation, and navigation—under a shared BasePlanning lifecycle and common runtime (please also see Reviewer g2ND’s Q2). As a result, extending PlanFactory is **modular** rather than framework-specific.
>
> 1. Decompose it into the four planning dimensions' code;
> 2. Write the corresponding planning prompt following the original paper or repository;
> 3. Implement framework-specific mechanisms as reusable tools within the runtime
>
> Based on our experience, implement and debug a new framework takes around 2–3 days. Most planning systems can be covered by the current abstraction, while the main exceptions are methods whose design conflicts with the underlying Flash-Searcher[1] infrastructure or assumes substantially different runtime behavior. We will add these engineering details to the appendix.
>
> ---
>
> ## Q3. Gain Source: Training or Search
>
> Thank you for this important point. For each query, TodoEvolve directly synthesizes a single planner and executes it, without test-time multi-candidate search. At the same time, our training pipeline does involve offline candidate exploration **during data construction**. The key claim is therefore not that search is absent, but that its benefits can be **distilled into a meta-planning policy** that generalizes at test time.
>
> To **isolate** training effect from search effect, we conducted a comparison on xBench-DS. We compare the Base and SFT models under Best-of-K test-time search against the final model with one-shot planner generation:
>
> | Model | Test-time Planner Candidates | Accuracy |
> | ----- | ---------------------------- | -------- |
> | Base        | 3 | 14.0 |
> | Base        | 5 | 19.0 |
> | SFT         | 3 | 38.0 |
> | SFT         | 5 | 44.9 |
> | SFT + IGPO  | 1 | 75.0 |
>
> Results show that gain can't be explained by sampling alone, even with multiple test-time candidates, Base/SFT models remain far below the trained meta-planner.
>
> ---
>
> ## Q4. Cognitive Impedance motivation
>
> Thank you for raising this point. Cognitive Impedance (CI) is used only to **rank multiple correct** plans during IGPO preference construction. If one candidate succeeds and the other fails, the successful one is always preferred. CI is used to penalize successful but inefficient plans (e.g., repeated errors or planning overhead).
>
> Formally
>
> $$
> CI = C_{tot} \cdot \exp \left( \lambda_1 N_{fail} + \lambda_2(1 - S_{stab}) + \lambda_3 \frac{C_{plan}}{C_{exec}} \right),
> $$
> where:
>
> - $C_{tot}$: Total trajectory cost
>
> - $N_{fail}$: The count of execution failures(e.g. runtime errors or failed tool calls).
>
> - $S_{stab}$: The execution smoothness, which is operationalized as $1 - \frac{\text{summary steps}}{\text{total steps}}$ means the execution proceeded directly without frequent summary or replanning interruptions.
>
> - $\frac{C_{plan}}{C_{exec}}$: A penalty to prevent the agent spend more tokens on generating the plan than it spends actually executing the task.
>
> We also performed a remove-one-term ablation:
>
> | Setting | Accuracy (%) | Avg Cost(\$) | Avg Time(s) |
> |---------|--------------|----------|----------|
> | Default CI | 70.00 |0.0167  |216.59  |
> | No-fail term ($\lambda_1 = 0$) | 62.33 | 0.0203 | 298.31 |
> | No-smoothness term ($\lambda_2 = 0$) | 67.67 | 0.0196 | 253.48 |
> | No-overhead term ($\lambda_3 = 0$) | 64.00 | 0.0182 | 272.37 |
>
> The default formulation gives the best trade-off; removing any term reduces accuracy and increases cost/time. The failure term has the largest effect, while smoothness and overhead provide complementary gains.
>
> ---
>
> ## Q5. Core concept issue
>
> Thank you for this kind feedback. We agree that these concepts are introduced abruptly and will reorganize them in the revision. Please also see our response to Reviewer ZQdY for the broader presentation clarifications.
>
> ---
> ### References:
> [1] Flash-Searcher: Fast and Effective Web Agents via DAG-Based Parallel Execution
>
> Sincerely,
>
> Authors of Submission 5400

---

> > ### Author Rebuttal · Reviewer_f6H5 · 2026-04-03
> >
> > The additional experiment is comprehensive.

---

> > > ### Author Response · Authors · 2026-04-03
> > >
> > > Dear Reviewer f6H5,
> > >
> > > Thank you very much for your thoughtful follow-up and for taking the time to review our additional experiments and clarifications. We sincerely appreciate your positive feedback and are very encouraged that our rebuttal has adequately addressed your concerns. Your comments have been highly valuable in helping us improve the clarity and presentation of the paper.
> > >
> > > Sincerely,
> > >
> > > Authors of Submission 5400

---

### Official Review · Reviewer_g2ND · 2026-03-13

**Soundness:** 3
**Presentation:** 2
**Significance:** 4
**Originality:** 3
**Overall Recommendation:** 5
**Confidence:** 3

**Summary:**

### Motivation

Previous LLM-based planning systems rely on static planning structures that lack the flexibility to adapt to novel problems.

The authors propose TodoEvolve, a sophisticated adaptive planning mechanism that synthesizes task-adaptive agentic planners (from linear plans to intricate graph structures) that dynamically update their states as execution progresses.

### Proposal

The architecture is split into an Agentic and a Planning layer with formal specifications for each. The previous methods can also be classified under the specifications.

Overall, the method consists of:

- **Unified Codebase:** Most previous planning paradigms encapsulate 4 components: *Topology* (structural organization of task decomposition), *Initialization* (how the task topology is instantiated), *Adaptation* (when and how the topology is revised), and *Navigation* (mechanism that issues executable directives to the acting agent). Therefore, propose a unified codebase, *PlanFactory*, for implementing/mixing and evaluating different planning mechanisms. A plan is defined using code as a class that inherits from the base class *BasePlanning* and implements the mentioned component functions. The Plan factory implements these specifications as tools from the previous papers (Table 1), and new plans can use these mechanisms as tools.
- **TodoEvolve:** A meta-planning paradigm that learns to instantiate and control planning structures on the fly. It employs an *Impedance‑Guided Preference Optimization* (IGPO) to train Todo‑14B, a 14B‑parameter meta‑planner capable of tailoring control flows across diverse scenarios and LLM backbones. Unlike standard alignment, which focuses on stylistic imitation, IGPO explicitly optimizes the meta-planner to maximize execution stability while minimizing computational overhead.

The TodoEvolve planning process (inference) can be summarized as:

1. For each user query $Q_i$, a specialized context $C_i$ is constructed containing the query, the meta planner’s role, documentation of the tools, and examples for reference.
2. Given the context $C_i$ as prompt, TodoEvolve generates a plan $P_i$.
3. Execute the plan $P_i$ using the PlanFactory runtime yielding the final answer $A_i$.

 For training, two datasets are constructed: SFT and IGPO data. SFT data contains plan examples that successfully execute, and pairs of plans (*win*, *lose*) for IGPO. The *win* plan in the IGPO data is evaluated using correctness and efficiency (Cognitive Impedance).

The TodoEvolve planner is then fine-tuned using two stages: SFT and IGPO.

### Results

**Training dataset** consists of: TaskCraft, MoNaCo, WebWalkerQA, and DeepSearchQA.

**Test datasets:** GAIA and XBench-DS. And constructed test splits from the train datasets.

- **Performance:** Up to $16.37\\%$ improvement (GPT5-Mini) over baseline Smolagents+GPT5-Mini on the GAIA benchmark. All leading pass@1 scores are from the TodoEvolve+Smolagents framework.
- **Generalization:** Performance improves using all LLM backbones: GPT5-Mini, Kimi K2, DeepSeek V3.2. Though using GPT5-Mini (best performing backbone), Flash-Searcher comes close with performance gaps between ($2\\%$ to $6\\%$) for all datasets.
- **Adaptivity:** Ability to generate task‑specific architectures that move beyond static templates, tailoring control flow to meet cognitive impedance and dependency requirements (Fig. 2).
- **Efficiency:** Performance gains do not come at the cost of more compute (Table 4).

Ablations compare 4 configurations on the GAIA level 2: Base model, SFT only, Zero-Shot (IGPO training without few-shot examples), and TodoEvolve. Results show significant performance drops when components are removed. But note that TodoEvolve also increases API cost, time, and average steps, which can be expected in some cases (zero-shot vs. few-shot).

**Compliance With Llm Reviewing Policy:**

Affirmed.

**Final Justification:**

All the concerns/clarifications have been addressed; I recommend acceptance.

**Key Questions For Authors:**

1. How is TodoEvolve integrated with the Smolagents framework?
2. How does the framework selection impact the performance?

**Limitations:**

A discussion of the approach's limitations is missing.
Maybe discussing any planning methods that fall outside the BasePlanning abstraction can help.

**Strengths And Weaknesses:**

### Strengths

1. Good abstraction of the planning mechanisms that encompasses many previous agentic planning methods.
2. The PlanFactory can be used for testing new plan methods.
3. Results show that the method helps in performance, generalization, and adaptivity.

### Weaknesses

1. Concrete examples of inputs/outputs of the meta planner and the agent inputs should be attached for clarity. And the prompts used (like for the LLM evaluator) should be added.
2. Some details about the PlanFactory are unclear. For example, the creation and usage of tools extracted in “Phase 1” of the dataset creation. A sub-section to Appendix A, expanding on the PlanFactory implementation details, can be helpful.

---

> ### Author Rebuttal · Authors · 2026-03-31
>
> Dear Reviewer g2ND,
>
> Thank you for your recognition and constructive feedback!
>
> ---
>
> ## Q1. End-to-End Pipeline
>
> Thank you for this suggestion. TodoEvolve does not directly produce the final answer. Instead, the meta-planner generates a task-specific planning system, which the downstream agent executes in PlanFactory. Due to space limits, we summarize the workflow here rather than provide a full query example.
>
> The **meta-planner’s input** is a **task-specific planning prompt** that includes:
> (1) User query;
> (2) Role-defining system prompt;
> (3) The abstract BasePlanning interface;
> (4) Generation objectives and formatting constraints;
> (5) Framework-level runtime specifications;
> (6) Three randomly sampled planning examples from seven decomposed planning frameworks;
> (7) Descriptions of the callable tools.
>
> The **meta-planner output** is not an answer, but a generated Python planning module together with a YAML configuration.
>
> The **execution-time agent input** consists of:
> (1) The original question;
> (2) The generated planning module;
> (3) The generated YAML configuration;
> (4) The callable tools and their descriptions.
> The agent then executes under this planner/configuration within PlanFactory to produce the final answer.
>
> The **judge prompt** is
>
> ```python
>     Please determine if the predicted answer is equivalent to the labeled answer.
>     Question:  {question}
>     Labeled Answer:  {gt_answer}
>     Predicted Answer: {pred_answer}
>     Are these answers equivalent?
>     The output should in the following json format:
>     {{
>     "rationale": "your rationale for the judgement, as a text",
>     "judgement": "your judgement result, can only be 'correct' or 'incorrect'"
>     }}
> ```
>
> In the revision, we will add a complete end-to-end example in the appendix.
>
> ---
> ## Q2. PlanFactory Tools
>
> Thank you for pointing this out. The “tools extracted in Phase 1” do not refer to ordinary task-level tools such as web search or code execution, but to **planning primitives abstracted** from the seven representative planning frameworks.
>
> PlanFactory decompose these systems and reorganize them into a modular design space through:
>
> (1) **Four-dimensional abstraction** - Topology, Initialization, Adaptation, and Navigation
> (2) Shared **BasePlanning** lifecycle under the same abstraction
> (3) A collection of standardized **tools**
> (4) Unified execution runtime
>
> To demonstrate how this architecture functions in practice, specifically regarding the "tools" extracted in Phase 1, we can look at a **concrete example**. To capture JoyAgent[1]'s core ensemble-style planning logic, we instantiated two primitive tools: EnsembleTool and VoteTool. EnsembleTool invokes different expert teams in parallel, while VoteTool aggregates their candidate answers via voting and critic-style synthesis. This reproduces JoyAgent’s original coordination mechanism within Planfactory.
>
> ---
>
> ## Q3. Framework selection
>
> Thank you for this insightful question! We would like to clarify that TodoEvolve and Smolagents[2] operate at **different layers of the system**. TodoEvolve acts as a meta-planner that synthesizes the high-level planning architecture $P$ , whereas Smolagents serves as the lower-level execution shell $M$ to handle environment interactions.
>
> Specifically, TodoEvolve generates a task-specific plan system that is instantiated within the downstream runtime; it does **not** replace the executor, but rather **provides the overarching planning layer** that orchestrates it. In other words, Smolagents defines how actions are executed, while TodoEvolve defines how planning structure is generated and updated.
>
> Regarding the framework's impact, our implementation follows the Flash-Searcher [3] setting and is built heavily upon Smolagents. Because TodoEvolve injects planning code into the runtime, porting it to other frameworks requires large engineering effort. However, our experiments already prove the efficacy, applying TodoEvolve yields up to a **24% gain** over the vanilla Smolagents baseline. Since standard frameworks share **similar execution spaces**, this gain mainly comes from TodoEvolve's dynamically optimized topologies rather than the specific execution shell.
>
> ---
>
> ## Q4. Limitations
>
> Thank you for this important suggestion. PlanFactory is intended as a **modular design space** for a broad class of planning systems through **topology**, **initialization**, **adaptation**, and **navigation**, rather than a complete abstraction of all agent orchestration paradigms. As a result, some highly stateful or memory-centric long-horizon agents may not be fully captured. We will clarify this limitation in the revision.
>
> ---
> ### References:
> [1] JoyAgent-JDGenie: Technical Report on the GAIA
>
> [2] smolagents: a smol library to build great agentic systems.
>
> [3] Flash-Searcher: Fast and Effective Web Agents via DAG-Based Parallel Execution
>
>
> Sincerely,
>
> Authors of Submission 5400

---

> > ### Author Rebuttal · Reviewer_g2ND · 2026-04-03
> >
> > We thank the authors for their clarifications. My concerns have mostly been addressed. The CI ablation, presented to another reviewer, further supports the soundness. If the authors add the missing details discussed above and other details required for reproducibility, I suggest accepting. A few minor questions and requests:
> >
> > 1. Can the authors add a formal interface/signature expected from each of the 4 functions required in a plan?
> > 2. How extensible can the approach be to incorporate plans that maintain a state during execution?
> > 3. Currently, I feel some pre-requisite knowledge is required for execution frameworks in general. A brief overview of the differences between the frameworks will be helpful.

---

> > > ### Author Response · Authors · 2026-04-04
> > >
> > > Dear Reviewer g2ND,
> > >
> > > Thank you for the thoughtful follow-up! We are encouraged that the main concerns are mostly resolved and appreciate these concrete suggestions.
> > >
> > > ---
> > >
> > > ## Q1: Interface of Planning Functions
> > >
> > > PlanFactory formalizes the planning lifecycle through four dimensions: **Topology**, **Initialization**, **Adaptation**, and **Navigation**, corresponding to how a planning system is structured, instantiated from the task, revised during execution, and translated into executable directives for the acting agent.
> > >
> > > | Dimension | Interface | Output |
> > > |---|---|---|
> > > | **Topology** | `topology(self, task: str) -> PlanningStep` | High-level plan structure |
> > > | **Initialization** | `initialize(self, task: str) -> PlanningStep` | Initial planning state / instantiated subgoals |
> > > | **Adaptation** | `adaptation(self, task: str, step: int, write_memory_to_messages: Callable[[Optional[List[ActionStep]], Optional[bool]], List[Dict[str, str]]]) -> SummaryStep` | Progress summary and updated planning guidance |
> > > | **Navigation** | `navigation(self, task: str, step: int, write_memory_to_messages: Callable[[Optional[List[ActionStep]], Optional[bool]], List[Dict[str, str]]]) -> ActionStep` | Action / tool-call decision with reasoning |
> > >
> > > The runtime infrastructure is from **Flash-Searcher**[1], where the lifecycle is recorded through three core step objects: `PlanningStep`, `SummaryStep`, and `ActionStep`.
> > >
> > > | Step class | Purpose | Key fields |
> > > |---|---|---|
> > > | `PlanningStep` | Stores plan-level outputs generated during topology / initialization | `plan`, `plan_think`, `plan_reasoning` |
> > > | `SummaryStep` | Stores adaptation outputs that summarize progress and revise the plan | `summary`, `summary_reasoning` |
> > > | `ActionStep` | Stores each executable navigation / execution step in runtime | `tool_calls`, `observations`, `error`, `action_output`, `action_reasoning`, `step_number`, `duration` |
> > >
> > > ---
> > >
> > > ## Q2: Extensibility to Stateful Plans
> > >
> > > As discussed above, PlanFactory already separates planning into four dimensions and can further support stateful execution through shared runtime state / memory.
> > >
> > > A concrete example is our integration of **FlowSearch**[2], whose original design requires maintaining and refining a dynamic knowledge flow during execution. It constructs a structured **knowledge flow** and iterates between a **Knowledge Collector and Refiner**, updating node states and graph structure dynamically. To reproduce this behavior, we introduced two framework-specific tools on top of the shared runtime:
> > >
> > > -  `Executor` tool which reads the current graph state and executes a selected node based on its tuple $(t_i, d_i, s_i, c_i)$.
> > > -  `Refine` tool which updates node status and knowledge context, and can further add/delete nodes and edges in the graph.
> > >
> > > More generally, our four-dimensional decomposition, together with each framework’s original plan/execution logic and corresponding runtime tools, can naturally support runtime re-planning and state updates. Frameworks requiring runtime structural adjustment can be incorporated by packaging persistent state and update logic as tools within the PlanFactory runtime.
> > >
> > > ---
> > >
> > > ## Q3: Framework differences overview
> > >
> > > To clarify the representative frameworks, we summarize their main planning styles below.
> > >
> > > - **Flash-Searcher**[1] and **OAgents**[3] are based on explicit task dependencies: the plan is decomposed into linked subtasks, and execution follows prerequisite relations.
> > > - **OWL**[4] and **AgentOrchestra**[7] are more hierarchical: a higher-level planner decomposes the objective and dispatches work to lower-level workers.
> > > - **JoyAgent**[6] is closer to an ensemble setting, where multiple expert teams work in parallel and their outputs are combined through voting, criticism, or synthesis.
> > > - **FlowSearch**[2] centers on a dynamically evolving knowledge structure that is expanded and refined during execution.
> > > - **Co-Sight**[5] is more verification-oriented, emphasizing cross-checking, inconsistency detection, and conflict resolution before advancing the plan.
> > >
> > > In short, these systems share a common planning abstraction but differ in how they organize, revise, and execute plans. Their concrete mapping to the four dimensions is provided in **Appendix A**. We will add a brief framework background overview.
> > >
> > > Faithfully,
> > >
> > > Authors of Submission 5400
> > >
> > > ---
> > > ### References:
> > > [1] Flash-Searcher: Fast and Effective Web Agents via DAG-Based Parallel Execution
> > >
> > > [2] FlowSearch: Advancing deep research with
> > > dynamic structured knowledge flow
> > >
> > > [3] OAgents: An Empirical Study of Building
> > > Effective Agents
> > >
> > > [4] OWL: Optimized Workforce Learning for General
> > > Multi-Agent Assistance in Real-World Task Automation
> > >
> > > [5] Co-Sight: Enhancing LLM-Based Agents via Conflict-Aware Meta-Verification
> > > and Trustworthy Reasoning with Structured Facts
> > >
> > > [6] JoyAgent-JDGenie: Technical Report on the GAIA
> > >
> > > [7] AGENTORCHESTRA: ORCHESTRATING
> > > HIERARCHICAL MULTI-AGENT INTELLIGENCE WITH
> > > THE TOOL-ENVIRONMENT-AGENT(TEA) PROTOCOL

---

### Decision · Program_Chairs · 2026-04-30

**Decision:**

Accept (regular)

**Comment:**

To address the limitation of fixed, manual planning structures in agentic systems, this paper presents TodoEvolve, a meta-planning paradigm that synthesizes and dynamically revises task-specific planning architectures, with the meta planner trained via Impedance-Guided Preference Optimization (IGPO),

The paper received 4 reviews with 3 Accepts (score 5) and 1 Reject (score 2). The three positive reviewers recognized that the paper addresses a well-motivated problem, offers a technically sound framework with a principled four-dimensional planning abstraction, and provides robust empirical evidence.

The other reviewer raised concerns primarily on framing and scope rather than about invalidating the core empirical contribution. During the rebuttal period, the authors clarified that the paper's objective is not to provide a universal abstraction of all possible agent system designs, but rather to focus on designing and learning the planning layer within agentic systems. This is well supported by the other three reviewers.

Thus, given the broad consensus among three out of four reviewers, I would recommend an acceptance decision.